# NOX4 functions as a mitochondrial energetic sensor coupling cancer metabolic reprogramming to drug resistance

Karthigayan Shanmugasundaram[1], Bijaya K. Nayak[1], William E. Friedrichs[1], Dharam Kaushik [2], Ronald Rodriguez[2] & Karen Block[1,3]

The molecular mechanisms that couple glycolysis to cancer drug resistance remain unclear. Here we identify an ATP-binding motif within the NADPH oxidase isoform, NOX4, and show that ATP directly binds and negatively regulates NOX4 activity. We find that NOX4 localizes to the inner mitochondria membrane and that subcellular redistribution of ATP levels from the mitochondria act as an allosteric switch to activate NOX4. We provide evidence that NOX4-derived reactive oxygen species (ROS) inhibits P300/CBP-associated factor (PCAF)-dependent acetylation and lysosomal degradation of the pyruvate kinase-M2 isoform (PKM2). Finally, we show that NOX4 silencing, through PKM2, sensitizes cultured and ex vivo freshly isolated human-renal carcinoma cells to drug-induced cell death in xenograft models and ex vivo cultures. These findings highlight yet unidentified insights into the molecular events driving cancer evasive resistance and suggest modulation of ATP levels together with cytotoxic drugs could overcome drug-resistance in glycolytic cancers.

[1] Department of Medicine, UT Health, San Antonio, TX 78229, USA. [2] Department of Urology, UT Health, San Antonio, TX 78229, USA. [3] South Texas Veterans Health Care System, San Antonio, TX 78229, USA. Correspondence and requests for materials should be addressed to K.B. (email: karen.block@va.gov)

Metabolic reprogramming, a hallmark of cancer, results from altered transcriptional, translational, and post-translational events, which together orchestrate a heightened activity within the cancer cell, in part, resulting in drug-resistance[1–3]. Molecular determination of aberrant oncogenic signaling events has been instrumental in the development of mechanism-based drug therapy. However, many promising drugs have yielded disappointing clinical outcomes due to activation of compensatory signaling pathways. Identifying underlying alternative signaling pathways and the functional interconnections that give rise to evasive resistance remain challenging in cancer research as uncloaking them requires identification of the existence that is concealed.

Metabolic reprogramming is characterized by reduced mitochondrial oxygen consumption with a shift in subcellular energy ATP production via aerobic glycolysis in the cytosol in lieu of the mitochondria through oxidative phosphorylation[4, 5]. The distinct molecular mechanisms coupling metabolic reprogramming to drug-resistance in cancer cells are unknown. However, the balance between reactive oxygen species (ROS) production and their neutralization via antioxidants, cumulatively known as redox homeostasis are intimately involved[6].

We and others have shown that the membrane bound NADPH oxidases of the NOX family are a major source of ROS in cancer[7–14]. Seven membrane-bound NOX catalytic isoforms, referred to as NOX1 to NOX5, dual oxidase 1 (DUOX1) and DUOX2 have been identified, each of which displays similar but distinct structural, biochemical, and subcellular localization characteristics. We were the first to show that NOX4 uniquely localizes to the mitochondria in various renal and endothelial cell types[8]. However, the mechanisms by which NOX4 is regulated within the mitochondrial compartment is unknown. Paradoxically, ROS produced by NOX4 has been linked to cancer cell survival through yet unidentified mechanisms[12, 15–18]. A role for NOX4 upstream or downstream of the metabolic switch has not been examined.

Renal cell carcinoma (RCC) most commonly arises from the loss of the von Hippel–Lindau (VHL) tumor suppressor gene and has the highest death rate among solid urological tumors. Despite surgery to remove the affected kidney (nephrectomy), ~30–40% of patients succumb to metastatic disease due to the lack of effective drug therapies and drug resistance.

Here we assessed the links between the NADPH oxidase isoform, NOX4, energetic metabolism, and cancer drug-resistance using VHL-deficient renal cancer cells as a model system.

## Results

**NOX4 directly binds ATP through a Walker A binding motif.** We examined the primary sequence of NOX4. Interestingly, we find that NOX4 harbors a putative, yet unexplored, Walker A, P-loop ATP/GTP binding motif (AXXXXGKT)[19] within amino acids 534–541 of the C terminus (Fig. 1a). Importantly, the Walker A motif is unique to NOX4 and is not found in other NOX isoforms (Fig. 1a). However, the Walker A motif is conserved in *Homo sapiens* (hNOX4), *Rattus norvegicus* (rNOX4), and *Mus musculus* NOX4 (mNOX4) (Fig. 1b). Together, this suggests a potential novel mechanism by which NOX4 may be allosterically regulated.

To delineate the functionality of the Walker A motif within NOX4, we examined the ability of NOX4 to directly bind ATP. To this end, we cloned human NOX4 sequences encoding amino acids 341–561 into the NusA vector system (NOX4$^{341–561}$), which enhances the solubility of recombinant protein expression. The NusA system harbors a 6xHIS tag at the C terminus for purification and characterization. In parallel, the Walker A lysine residue described as non-dispensable for ATP binding was mutated by site-directed mutagenesis to alanine (K540A, NOX4$^{341–561MUT}$) (Supplementary Fig. 1a)[20–22]. Recombinant (r) NusA fusion proteins (rNOX4$^{341–561}$ and rNOX4$^{341–561MUT}$) were generated, purified, and characterized. Coomassie blue staining shows the fusion proteins were soluble and pure and migrated at a predicted molecular weight of ~79 kDa (NusA, 54.8 kDa, and NOX4 $^{341–561}$, 24.2 kDa) (Supplementary Fig. 1b). BSA was included for protein quantitation. To demonstrate specificity of the purified recombinant proteins, increasing equal amounts of indicated recombinant proteins were resolved on SDS-polyacrylamide gel (PAGE) and western blot analysis was performed independently using anti-NOX4 and anti-6xhis antibodies (Supplementary Fig. 1c, d, respectively).

To determine whether ATP directly binds NOX4, equal amounts (1 µg) of rNOX4$^{341–561}$ were incubated with increasing doses (0.125–1.0 µCi) of α-$^{32}$P-labeled ATP and blotted onto 0.45-micron nitrocellulose as described[21]. The membranes were washed to best rid of unincorporated [α-$^{32}$P]-ATP and the bound protein with incorporated [α-$^{32}$P]-ATP was quantitated by scintillation. Figure 1c shows a dose-dependent increase of [α-$^{32}$P]-ATP binding to rNOX4$^{341–561}$ and expressed as dpm to background where the background is contributed by unwashed α-$^{32}$P sticking to the membrane.

To characterize specificity, we pre-incubated cold ATP prior to incubation with [α-$^{32}$P]-ATP for competition using decreasing concentrations of rNOX4$^{341–561}$. Figure 1d shows a dose-dependent incorporation of [α-$^{32}$P]-ATP to rNOX4$^{341–561}$ (1.0 µg vs. 0.5 µg), which was efficiently abolished by pre-incubation with cold ATP. We then examined the ability of rNOX4$^{341–561MUT}$ to bind [α-$^{32}$P]-ATP. In vitro binding assay was performed using equal amounts (1 µg) of rNOX4$^{341–561}$ and rNOX4$^{341–561MUT}$. Figure 1e shows the rNOX4$^{341–561}$ readily binds [α-$^{32}$P]-ATP, whereas rNOX4$^{341–561MUT}$ was unable to bind [α-$^{32}$P]-ATP. As an independent approach, we evaluated rNOX4$^{341–561}$ and rNOX4$^{341–561MUT}$ binding affinity to ATP using surface plasmon resonance (SPR)-based technology (Pioneer SensiQ, Sinsiq Technologies, performed by Affina Biotechnologies). Equal concentrations of rNOX4$^{341–561}$ and rNOX4$^{341–561MUT}$ proteins were immobilized by covalent amine-directed immobilization on a Ni-NTA chip surface using the Ni-His tag interaction as a pre-condensation step. The results show a rise in signal (RU) peaking at 80–100s for rNOX4$^{341–561}$ when ATP was passed over the flow channel, whereas rNOX4$^{341–561MUT}$ showed no binding/interaction signal (Fig. 1f top and bottom panels, respectively). Unfortunately, a negative SPR signal is present in Fig. 1f, bottom. Although it is not easy to identify the exact cause of such experimental artifact (buffer mismatch, volume exclusion, non-specific matrix interaction, and/or non-specific reference interaction), it is clear that there is an important difference in the interaction of rNOX4$^{341–561WT}$ and rNOX4$^{341–561MUT}$ with ATP. In summary, although in a very qualitative manner, our SPR analysis shows that NOX4 binds ATP directly through the Walker A binding motif, within amino acids 534–541 of the C terminus.

Physiological levels of intracellular ATP range between 1 and 10 mM. To determine the effects of ATP on NOX activity, we prepared RCC cell homogenates and added increasing concentrations of exogenous ATP. We find a dose-dependent reduction of NADPH-dependent superoxide generation with increasing ATP concentrations (2.0–5.0 mM) (Fig. 1g), whereas incubation of nucleoside adenosine diphosphate (ADP) or guanosine triphosphate (GTP) showed no change on NOX activity in RCC 786-O and A498 homogenates (Fig. 1h, i, respectively).

**Mitochondrial NOX4 is regulated by ATP.** We and others have previously demonstrated NOX4 is expressed and upregulated in

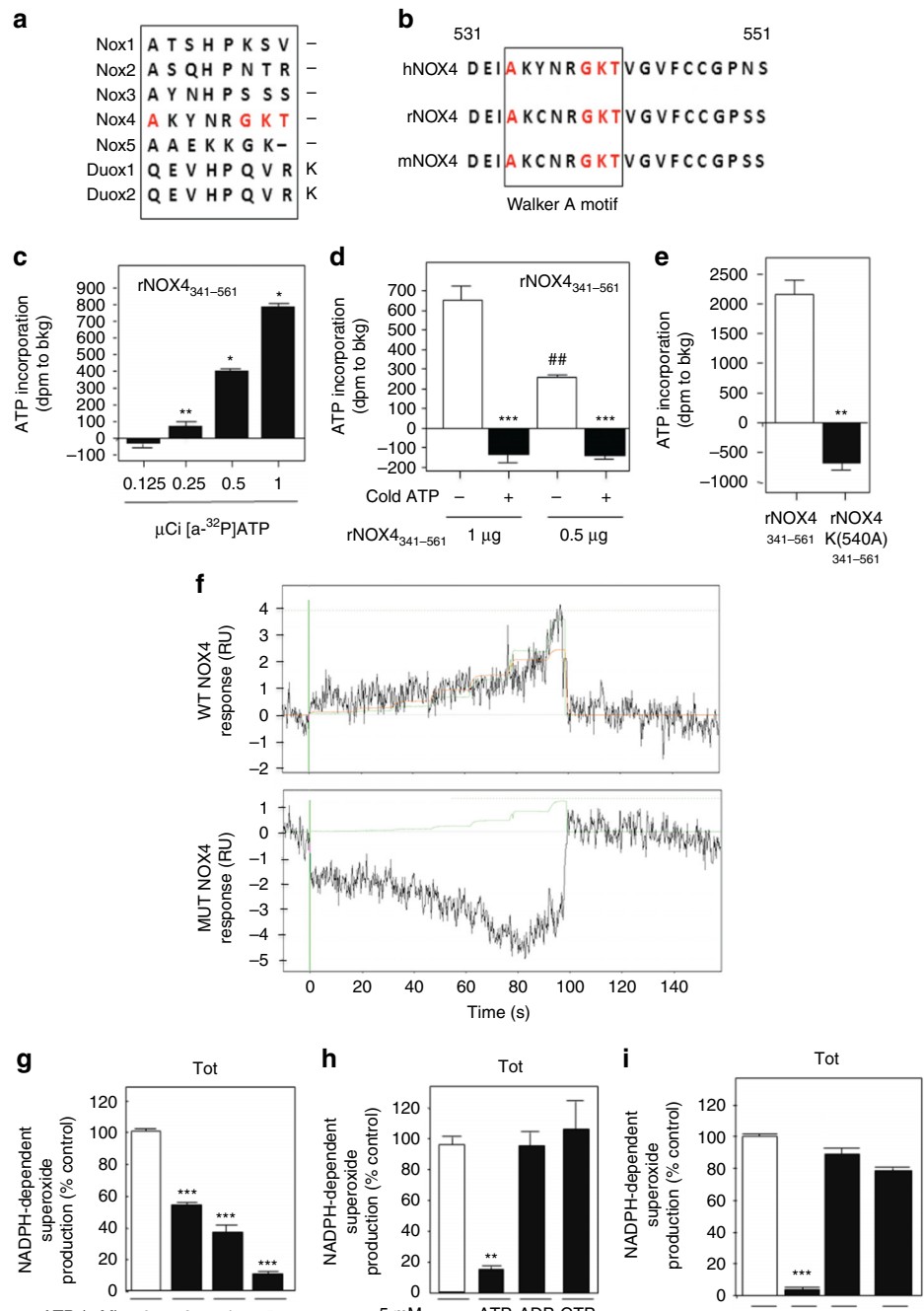

**Fig. 1** ATP directly binds NOX4 and negatively regulates NOX4 activity. **a** Alignment of the human NOX isoforms; NOX1–5, DUOX 1, and DUOX 2 shows a Walker A, ATP-binding motif (A/GXXXGKT/S) uniquely within the NOX4 isoform. **b** The Walker A ATP-binding motif is located at amino acids 534–541 conserved among Homo sapiens (hNOX4), Rattus norvegicus (rNOX4), and Mus musculus (mNOX4). **c** In vitro ATP-binding assay was performed using equal amounts (1 μg) of recombinant WT NOX4$^{341-561}$ incubated with increasing doses (0.125-1.0 μCi) of $^{32}$P-labeled ATP ([α-$^{32}$P]-ATP) and blotted onto 0.45-micron nitrocellulose, washed, and counted by scintillation as described in Methods. The results are presented as counts (dpm) to background (bkg) and are representative of two independent experiments of an $n = 6$ per sample. **d** In vitro ATP-binding assay was performed as in **c**; however, excess cold ATP was incubated with 0.5 or 1.0 μg recombinant WT NOX4$^{341-561}$ protein prior to [α-$^{32}$P]-ATP incubation. The results are presented as counts (dpm) to background (bkg) and are representative of two independent experiments of an $n = 6$ per sample. **e** In vitro ATP-binding assay was performed using 1 μg recombinant WT NOX4$^{341-561}$ and 1 μg MUT NOX4$^{341-561}$, K540A protein. **f** Surface plasmon resonance-based technology was employed to evaluate ATP binding to recombinant WTNOX4 and the ATP MUTNOX4 as described in Methods. The results are presented as counts (dpm) to background (bkg) and are representative of two independent experiments of an $n = 6$ per sample. **g** Increasing doses of ATP (2–5 mM) were incubated in total homogenates prepared from RCC 786-O cells. NADPH-dependent superoxide generation (NOX activity) was measured by enhanced chemiluminescence as described in Methods. **h** NOX activity was examined in RCC 786-O or **i** A498 total homogenates pre-incubated with 5.0 mM ATP, ADP, or GTP on ice for 30 min. Unless otherwise noted, the results are from at least three independent experiments and expressed as the means using one-way ANOVA with Tukey's post hoc test where ±S.E.M. *$p < 0.05$, **$p < 0.01$, ***$p < 0.001$ compared to control

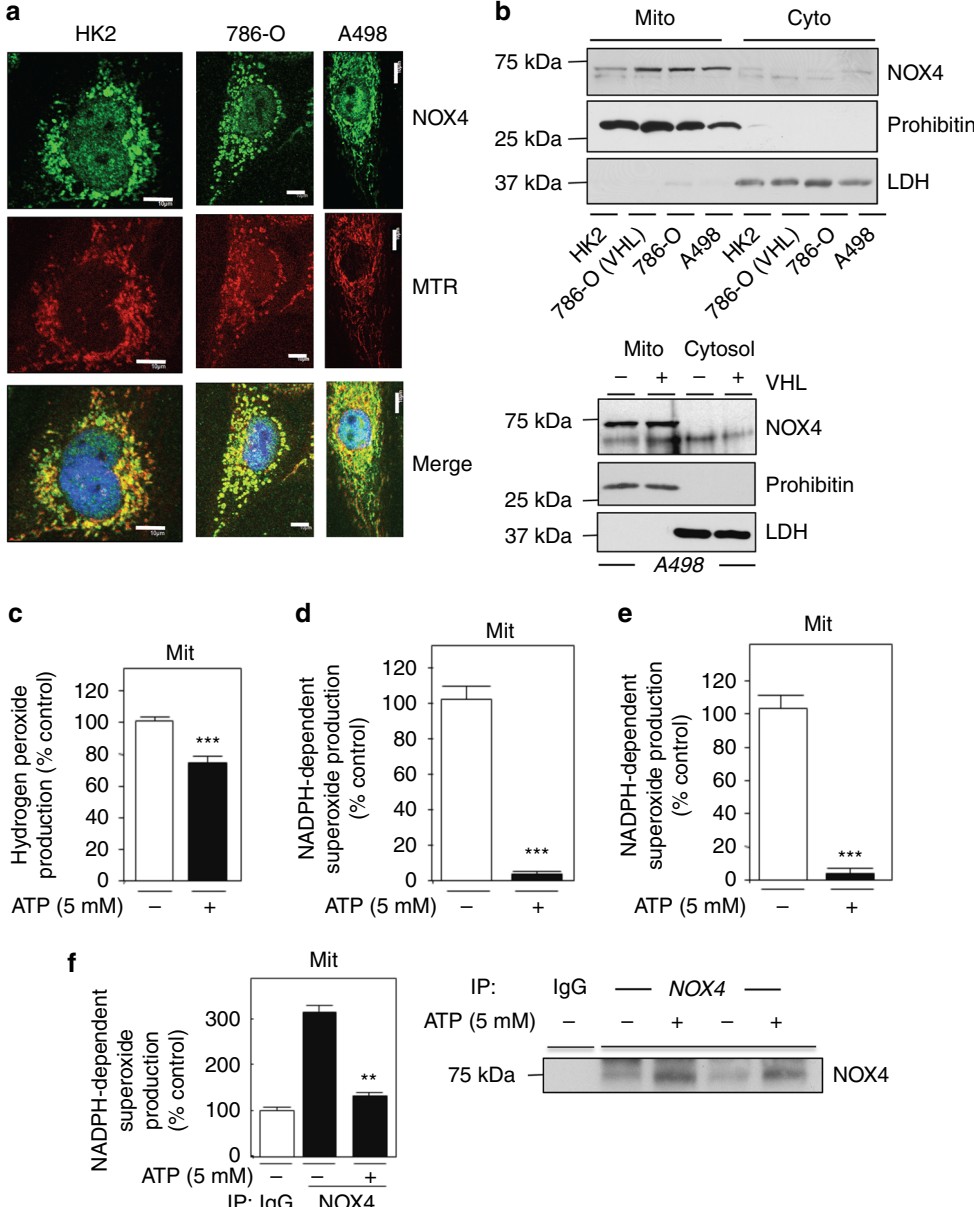

**Fig. 2** NOX4 localizes to the mitochondrial compartment and is inhibited by ATP. Subcellular localization of NOX4 was assessed in normal renal epithelial cells, HK2, and von Hippel Lindau (VHL)-deficient renal carcinoma cells (RCCs), 786-O and A498 by confocal microscopy as described[8]. **a** Indicated renal cells were labeled with mitotracker red (MTR), fixed, and stained with NOX4 (Novus49) antibody followed by FITC-linked donkey anti-rabbit secondary antibody. Nuclei were counter stained with DAPI. Merge in yellow shows colocalization. Scale bar 10 μM. **b** HK2, 786-O (with or without VHL), and A498 cells (upper panel) or A498 with or without VHL (lower panel) were subjected to pierce kit fractionation as per the manufacturer's instructions to yield mitochondrial or cytosolic fractions. NOX4 expression was assessed by western blot analysis. Prohibitin was used as a marker for mitochondria, whereas lactate dehydrogenase (LDH) was used as a marker for cytosol. **c** Hydrogen peroxide was measured in mitochondrial fractions as prepared, detailed in Methods, and was incubated with (+) or without (−) exogenous 5 mM ATP using Amplex Red reagent. The results are from two independent experiments with an $n = 4$ in each experiment. NOX activity was measured in crude mitochondrial homogenates with (+) or without (−) 5 mM ATP in both **d** 786-O and **e** A498 cells. **f** NOX4 was immunoprecipitated with NOX4 antibodies or IgG control independently using equal volume crude mitochondrial fraction preparation. The immunoprecipitates were washed and pre-incubated for 30 min on ice with 5 mM ATP, or buffer control, and NADPH-dependent superoxide generation was examined using the immunoprecipitates by enhanced chemiluminescence and normalized to IgG control as described in ([8]). The results are from at least three independent experiments and expressed as the means using one-way ANOVA with Tukey's post hoc test where ±S.E.M. *$p < 0.05$, **$p < 0.01$, ***$p < 0.001$ compared to control

cultured renal carcinoma cells (RCC) and human RCC tissue compared to normal controls[7, 8, 10–13, 23]. We evaluated the subcellular localization of NOX4 by confocal microscopy in RCC cells deficient of the VHL tumor suppressor gene (786-O and A498) and normal renal epithelial (HK2) cells. Indirect immunofluorescence was performed using NOX4 primary antibody

and FITC-conjugated anti-rabbit secondary antibody. NOX4 showed a precise spatial spaghetti-like organizational pattern resembling putative mitochondrial localization (Fig. 2a). To define this observation, mitochondria were stained in parallel using MitoTracker Red (MTR), a red-fluorescent dye that stains mitochondria in live cells. Figure 2a and Supplementary Fig. 2

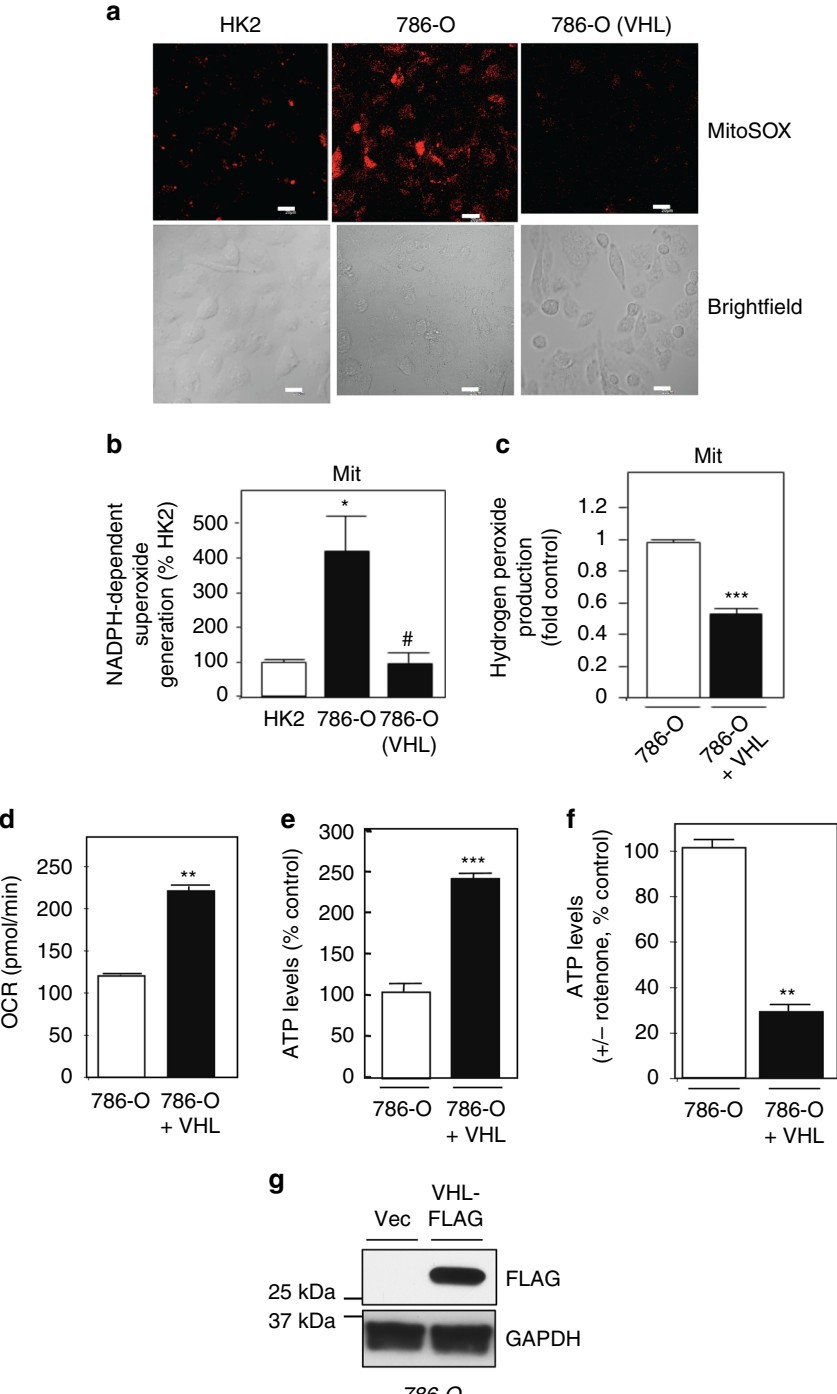

**Fig. 3** Redistribution of mitochondrial ATP levels correlates with NOX activation. **a** Superoxide generation within the mitochondrial compartment was examined in normal HK2 and RCC 786-O with and without VHL using MitoSOX Red fluorescence and confocal microscopy. Bright-field microscopy is shown (lower panel), scale bar 20 μM. **b** NADPH-dependent superoxide generation was examined as a readout of NOX activity in the mitochondrial fraction prepared from normal HK2 and RCC 786-O with and without VHL. **c** Hydrogen peroxide was measured in mitochondrial fractions of RCC 786-O with (+) or without (−) VHL cells using Amplex Red reagent. The results are from at least three independent experiments and expressed as the means where ±S.E.M. *$p < 0.05$ compared to HK2 control, ±S.E.M. #$p < 0.01$ or ***$p < 0.001$ compared to 786-O alone. **d** OCR, a measurement of oxygen consumption was measured in live RCC 786-O cells with and without VHL using the Seahorse XF instrument. The data are expressed in pmol/min are representative of three independent experiments with an $n = 4$ for each experiment and expressed as the means using one-way ANOVA with Tukey's post hoc test where ±S.E.M. **$p < 0.05$ compared to RCC 786-O without VHL. **e** Total ATP levels were measured in RCC 786-O cells with or without VHL. **f** RCC 786-O cells with or without VHL were incubated, 30 min or not with rotenone (1 μM) and total ATP levels were measured as in **e**. **g** Successful VHL overexpression was examined in TCL by western analysis using FLAG antibodies. GAPDH was probed as control. The reduction of ATP by rotenone was subtracted from buffer control and expressed as % control. The results are from at least three independent experiments and expressed as the means using one-way ANOVA with Tukey's post hoc test where ±S.E.M. **$p < 0.01$ and ***$p < 0.001$ compared to control

demonstrate overlay between NOX4 staining and MTR as indicated by the yellow coloration and suggests NOX4 localizes to the mitochondria in renal proximal tubular cells.

To independently verify NOX4 expression and subcellular localization within the mitochondrial compartment, we biochemically fractionated HK2, or RCC 786-O and A498 cells with (+) or without (−) VHL[24] into cytosolic and mitochondrial fractionations using the Pierce kit[8]. NOX4 expression was analyzed by immunoblotting in the aforementioned fractions. NOX4 is detected in the mitochondrial fraction, but not in the cytosolic fraction (Fig. 2b upper and lower panel). Notably, NOX4 expression is higher within this compartment in RCC cells, with or without VHL, compared to normal HK2 cells (Fig. 2b upper and lower panel). Efficient fractionation was evaluated using Prohibitin as a mitochondria marker and lactate dehydrogenase (LDH) as a cytosolic marker.

Mitochondria are composed of two phospholipid bilayer membranes. As NOX4 is a membrane bound protein, we biochemically characterized the differential subcompartmentalization of NOX4 to the inner or outer mitochondrial membrane. To this end, mitochondria were purified and exposed to the broad-spectrum protease, protease K over time. VDAC, an outer mitochondrial protein was present at T0, which was markedly reduced by over 60% at 15 min in the presence of Proteinase K (Supplementary Fig. 3a, b). On the other hand, Cox VI, a known inner mitochondrial protein was not affected by proteinase K digestion over the 15-min time digestion (Supplementary Fig. 3a, b). The stability of NOX4, in the presence or absence of proteinase K, was examined in parallel by western blot analysis using two independent NOX4 antibodies as described[8]. We find that NOX4 expression was not altered in the presence of proteinase K (Supplementary Fig. 3a, b). Together our data suggest that NOX4 resides in the inner mitochondrial membrane or on the inside of the outer mitochondrial membrane.

We next examined the functional contribution of NOX4 in the production of ROS within the mitochondrial compartment. NOX4 was stably silenced using small hairpin loop RNA (shRNA) to generate two independent single-cell knockdown clones (shNOX4-[1] and shNOX4-[2]) or transiently using small inhibitory RNA (siRNA) in VHL-deficient 786-O and A498 cells, respectively. Downregulation of NOX4 mRNA (Supplementary Fig. 4a) and protein expression (Supplementary Fig. 4b, c) compared to respective controls were verified.

Detection of superoxide within the mitochondrial compartment of live cells was analyzed using the MitoSOX red reagent[8]. Using confocal microscopy, we find that shRNA- and siRNA-mediated knockdown of NOX4 reduces ROS generation within the mitochondrial compartment in 786-O and A498 cells compared to control transfected cells (Supplementary Fig. 4d). In support of this finding, biochemical characterization of NOX activity[7, 8, 10, 11, 13] showed a marked reduction of NADPH-dependent superoxide production and hydrogen peroxide[25] in isolated mitochondrial fractions silenced of NOX4 compared to controls (Supplementary Fig. 4e, f).

As our data show NOX4 localizes to the mitochondria and is a source of ROS within this compartment, we prepared mitochondrial fractions from 786-O and A498 cells and examined NOX activity and hydrogen peroxide in the presence or absence of ATP. In support of our data, we find reduction of hydrogen peroxide levels and NADPH-dependent superoxide generation when ATP was incubated in isolated mitochondrial fractions as measured by Amplex red and enhanced chemiluminescence respectively (Fig. 2c, d (786-O), and Fig. 2e (A498)). To selectively determine if the reduction of NOX-dependent ROS by ATP is directly contributed by NOX4 NOX4 or IgG control was immunoprecipitated from mitochondrial fractions and NOX

activity was assessed in the presence (+) or absence (−) of exogenously added ATP on the immunoprecipitate. Figure 2f left panel shows NOX4-derived ROS is reduced in the presence of ATP compared to buffer control from two-independent assays. Successful immunoprecipitation of equal amounts of NOX4 (anti-NOX4) or control (IgG) used for the assays were assessed by western blot analysis in the immunoprecipitates by boiling the beads and resolving on SDS-PAGE (Fig. 2f right panel).

**Mitochondrial NOX4 is a novel energetic sensor**. We assessed the links between alterations in intracellular ATP levels, mitochondrial NOX4 activity, and ROS levels within the mitochondria using independent approaches. ROS within the mitochondria was assessed using MitoSOX in normal HK2, 786-O, and A498 cells with or without VHL. Confocal microscopy shows enhanced ROS production within the mitochondrial compartment in VHL-deficient cells, compared to normal HK2 cells and RCC cells where VHL was reintroduced (Fig. 3a, Supplementary Fig. 5a). This finding was paralleled when NOX activity (Fig. 3b, Supplementary Fig. 5b) and amplex red (Fig. 3c) were examined in isolated mitochondria.

Metabolic reprogramming is characterized by reduced mitochondrial oxygen consumption with a shift in subcellular energy ATP production via aerobic glycolysis in the cytosol in lieu of the mitochondria through oxidative phosphorylation[4, 5]. Importantly, we find reintroduction of VHL markedly enhances oxidative phosphorylation (OCR) as measured in live cells by Seahorse in 786-O and A498 cells (Fig. 3d, Supplementary Fig. 5c) concomitant with an increase in ATP levels (Fig. 3e, Supplementary Fig. 5d), which was reduced by pre-incubation with the mitochondrial complex I inhibitor, rotenone (Fig. 3f, Supplementary Fig. 5e). Successful VHL-FLAG transfection and expression was verified by FLAG western analysis. GAPDH was probed as a loading control (Fig. 3g, Supplementary Fig. 5f).

Culturing mammalian cancer cells with galactose as the source of sugar in lieu of glucose blocks aerobic glycolysis and shunts energy reliability to mitochondrial respiration[26]. As an independent approach, we cultured 786-O cells with glucose or galactose containing media for 24 h in the presence or absence of rotenone and measured ATP levels. In parallel plates, mitochondrial fractions were prepared and NOX activity was examined. We find that ATP levels were increased ~40% in RCC cells cultured in galactose, which was inhibited by rotenone, whereas glucose had no effect (Supplementary Fig. 6a). The increase of ATP levels in the mitochondria from galactose-treated cells was concomitant with a paralleled reduction of NOX activity (Supplementary Fig. 6b).

**NOX4 couples energy metabolism to drug-resistance**. We next sought to understand the biological significance of mitochondrial NOX4 activation. RCC cells are characteristically resistant to drug-induced apoptosis. To determine whether NOX4 mediates drug resistance in RCC cells, we treated 786-O cells (stably transfected with shVector or shNOX) or A498 cells (transiently transfected with Scr or siNOX4) with buffer alone or etoposide. Apoptosis was analyzed by Annexin V staining and flow cytometry. Importantly, RCC cells exposed to etoposide showed no, or minimal, enhancement of cell death compared to buffer alone, whereas RCC cells silenced of NOX4 demonstrate a marked increase of etoposide-induced cell death (Fig. 4a, Supplementary Fig. 7, respectively). The variability of cell death noted between 786-O and A498 cells is likely due to the added stress of transient transfection of A498 cells. In support, PARP cleavage was enhanced in NOX4 knockdown cells treated with etoposide (Fig. 4b). Actin was probed as a loading control.

We find herein that NOX4 predominately localizes to the mitochondria; however, NOX4 has been detected in other subcellular localizations of different cell types such as the endoplasmic reticulum, nucleus, and plasma membrane[8, 27–29]. To examine the role of ROS from the mitochondrial compartment to mediate drug resistance, we treated VHL-deficient cells

with mitotempol, a mitochondrial superoxide scavenger. We find 786-O cells pre-incubated (6 h) with mitotempol, sensitizes the cells to drug-induced cell death compared to drug-treated or mitotempol alone (Fig. 4c). To determine whether alterations in mitochondrial ATP levels are associated with drug-resistance, we examined sensitivity of drug-induced apoptosis in VHL-deficient

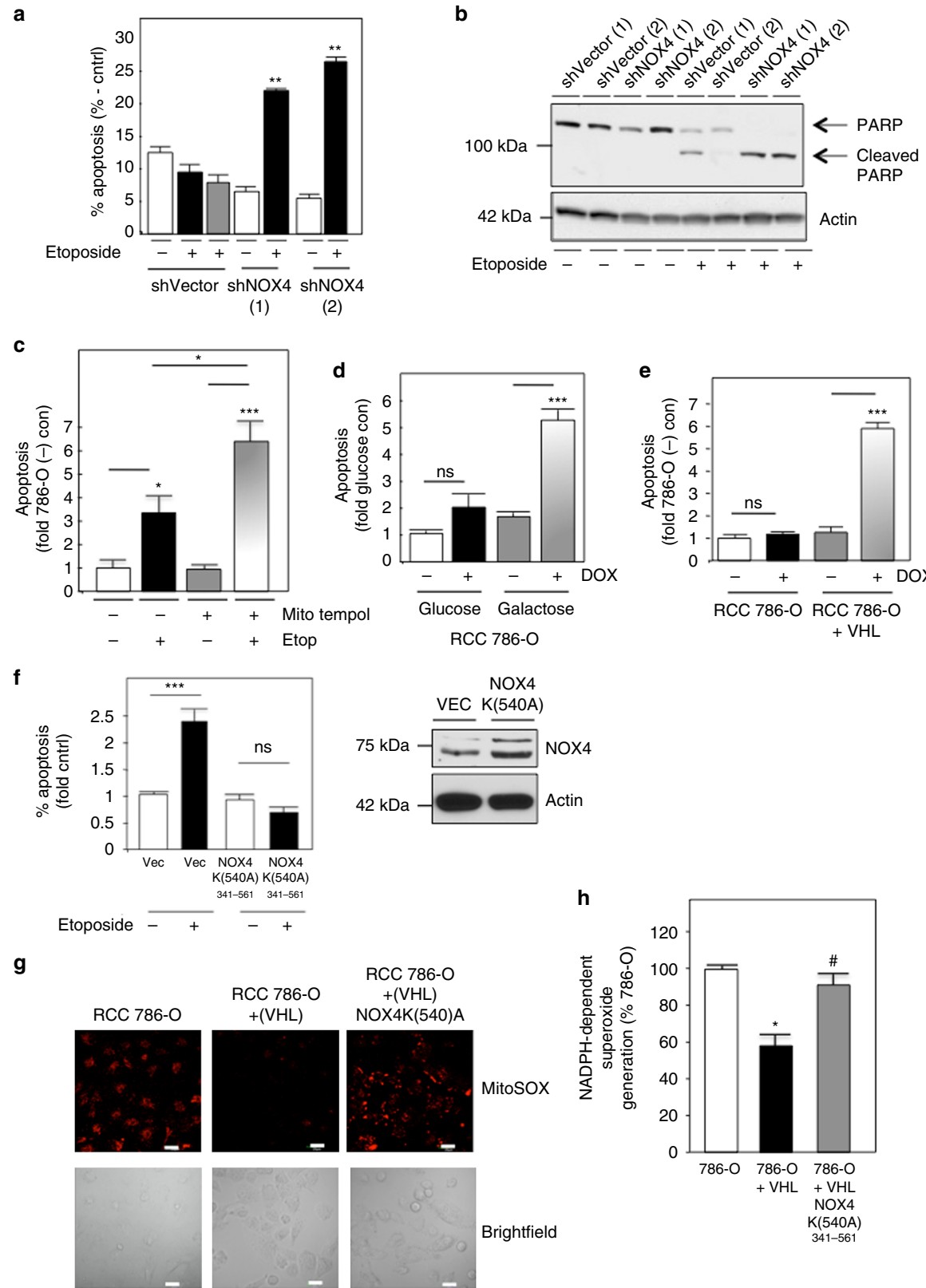

cells with VHL reintroduced and in 786-O cells cultured in glucose or galactose where we find the latter of each condition enhances ATP levels in the mitochondria concomitant with reduced NOX4 activity (Supplementary Fig. 6a, b). We find the indicated cytotoxic drugs readily induced cell death in galactose-treated and VHL add back RCC cells compared to VHL-deficient cells grown in normal glucose (Fig. 4d, e). To definitively show ATP-dependent regulation of NOX4 is necessary and sufficient to mediate drug resistance, we generated a NOX4 ATP mutant that does not bind ATP (K540A) (NOX4$^{K540A}$) in a mammalian expression vector. 786-O (+VHL) were transfected with vector control or NOX4$^{K540A}$ and the cells were subsequently exposed to etoposide (12–14 h). We find 786-O cells (+VHL/Vec/Etop) enhanced cell death compared to 786-O cells (+VHL/Vec/Buff). On the other hand, 786-O cells (+VHL/NOX4$^{K540A}$/Etop) did not demonstrate drug-induced cell death compared to control 786-O cells (+VHL/NOX4$^{K540A}$/Buff) (Fig. 4f left panel). Successful expression of NOX4 ATP mutant is demonstrated (Fig. 4f right panel). Figure 4g shows the NOX4 K(540A) mutant retains ROS generation as assessed by MitoSOX and Fig. 4h NOX activity compared to 786-O cells expressing VHL, suggesting it acts as a dominant negative.

**NOX4 mediates drug resistance through PKM2.** The downstream target of NOX4, which mediates drug resistance in cancer cells is unknown. Pyruvate kinase catalyzes the last step within glycolysis. The aberrant expression and/or post-translational modification of PKM2 have been linked to cancer cell growth and survival through complex, yet unclear mechanisms[30–33]. To determine whether PKM2 is a novel mediator of drug resistance in RCC cells, we silenced PKM2 in 786-O and A498 cells using siRNA. Importantly, PKM2 knockdown sensitizes RCC cells to drug-induced cell death, resembling NOX4 (Fig. 5a, Supplementary Fig. 8a, respectively). PARP cleavage was noted in siPKM2 cells treated with etoposide (Fig. 5b, Supplementary Fig. 8b, respectively). Successful PKM2 knockdown was verified by western blot (Fig. 5c, Supplementary Fig. 8c, respectively).

As we find siRNA-mediated knockdown of PKM2 sensitizes RCC cells to drug-induced cell death, we examined PKM2 expression in NOX4-silenced RCC cells. Interestingly, we find PKM2 levels were reduced in shNOX4 clones compared to shVec controls (Fig. 5d left and right panel). On the other hand, PKM1 was not altered (Fig. 5e). Acetylation of PKM2 mediates its destruction through a lysosomal mediated degradation pathway[34]. We examined PKM2 acetylation status in VHL-deficient cells stably transfected with shVector and shNOX4. PKM2 was immunoprecipitated and the immunoprecipitates were resolved

on an SDS-PAGE gel and probed using acetylation antibodies. We find PKM2 acetylation (58 kDa), is higher in NOX4 knockdown cells compared to shVector control (Fig. 6a). To determine whether NOX4 contributes to PKM2 stability through an acetylation-dependent, lysosomal-degradation mechanism, we treated shNOX4 786-O and shVector control cells with 3-methyladenine (3-MA), to block lysosomal degradation[35, 36]. Figure 6b left panel shows PKM2 expression is markedly higher in shNOX4 cells treated with 3-MA (+) compared to buffer control (−). Quantitation of PKM2 expression is presented (Fig. 6b right panel).

The effects of 3-MA on etoposide-induced cell death in shNOX4 knockdown cells were assessed. We find that shNOX4 sensitizes VHL-deficient cells to etoposide-induced cell death and that pre-treatment with 3-MA reduces this effect (Fig. 6c). In parallel, PARP cleavage was examined by western blot analysis. Supplementary Fig. 9a, b shows less PARP cleavage in etoposide treated VHL-deficient shNOX4 clones pre-treated with 3-MA. PKM2 is post-translationally acetylated by PCAF at K305 and p300 at K433, which mediates lysosomal degradation or nuclear localization respectively. To validate our findings that acetylation-dependent degradation of PKM2 is a mechanism that regulates drug-induced cell death in RCC, we examined cell death in RCC cells silenced PCAF or p300 using siRNA. We find that siRNA knockdown of PCAF but not p300 sensitizes RCC cells to etoposide-induced cell death compared to control (Fig. 6d). PKM2 levels were stabilized in siPCAF cells compared to sip300 and scrambled control (Fig. 6e). Successful knockdown of PCAF and p300 are shown in Fig. 6e. GAPDH was used as loading control. To definitively demonstrate PKM2 stabilization, through escaping acetylation- and lysosomal-mediated degradation, drives drug-resistance in RCC cells, we generated an acetylation mutant of PKM2 by mutating lysine 305 (the PCAF-acetylation site) to arginine (K305R) (PKM2$^{K305R}$) as described by Lv et al.[34]. 786-O cells stably silenced of NOX4 were transfected with Vector control or PKM2$^{K305R}$ and subsequently exposed to etoposide. As supported by our earlier data, shNOX4 786-O cells transfected exposed to etoposide induced cell death. However, 786-O shNOX4 cells transfected with PKM2$^{K305R}$ blocked drug-induced cell death (Fig. 6f upper panel). Successful expression of Flag-tagged PKM2 acetylated mutant was confirmed (Fig. 6f lower panel). Importantly, overexpression of NOX4$^{K540A}$ in shNOX4 knockdown RCC cells resulted in stabilization of PKM2 (Fig. 6g). In support of these findings and similar to our finding in Fig. 4f that NOX4$^{K540A}$ blocks drug-induced apoptosis in RCC cells re-expressing VHL, we find that overexpression of NOX4$^{K540A}$ in RCC cells re-expressing VHL stabilizes PKM2

**Fig. 4** Inhibition of NOX4 sensitizes RCC cells to drug-induced cell death. **a** shVector or shNOX4 786-O stable cell lines were incubated (+) or not (−) with etoposide. Apoptosis was analyzed by Annexin V staining and flow cytometry. The results are from at least three independent experiments and ±S.E.M. **p < 0.01 compared to etoposide-treated shVector control. **b** PARP cleavage was assessed by western blot analysis in parallel using lysates from **a**. Actin was used as the loading. **c** 786-O cells were pre-treated (6 h) with 250 nM of MitoTEMPOL and exposed to etoposide 12–14 h. Apoptosis was analyzed. The results are from at least three independent experiments and ±S.E.M. *p < 0.05 etoposide-treated cells compared to buffer and ***p < 0.001 is MitoTEMPOL plus etoposide compared to MitoTEMPOL alone. **d** 786-O cells were grown in complete media with glucose or galactose (20 h) then exposed to doxorubicin. Apoptosis is expressed ±S.E.M. ***p < 0.001 is compared with non-treated galactose. NS represents non-significance. **e** 786-O cells with or without VHL were exposed to doxorubicin (1 μg/ml). Apoptosis was measured as in **a**. The results are from three independent experiments where ±S.E.M. ***p < 0.001 is compared with 786-O cells (+) VHL. **f** (Left panel) 786-O (+VHL) was transfected with Vector or ATP mutant NOX4-(K540A). After 24 h, the cells were exposed to etoposide. Apoptosis was assessed. The results are from two independent experiments and the results are expressed as % apoptosis normalized to fold control using one-way ANOVA with Tukey's post hoc test where ±S.E.M. ***p < 0.001 Vector (+) etoposide compared to Vector control without (−) etoposide. NS represents non-significance. (Right panel) Overexpression of NOX4 mutant was verified. Actin was used as a loading control. **g** Superoxide within the mitochondria (mitoSOX) or **h** NADPH-dependent superoxide generation was assessed in 786-O or 786-O with VHL plus or minus NOX4$^{341-561MUT}$. Representative fluorescent and bright-field images obtained by confocal fluorescence microscopy are shown, Scale bar 20 μM. Results are from two independent experiments and expressed as percent control (786-O) where ±S.E.M. *p < 0.05 786-O and #p < 0.05 786-O + VHL. Statistics are expressed as the means using one-way ANOVA with Tukey's post hoc test

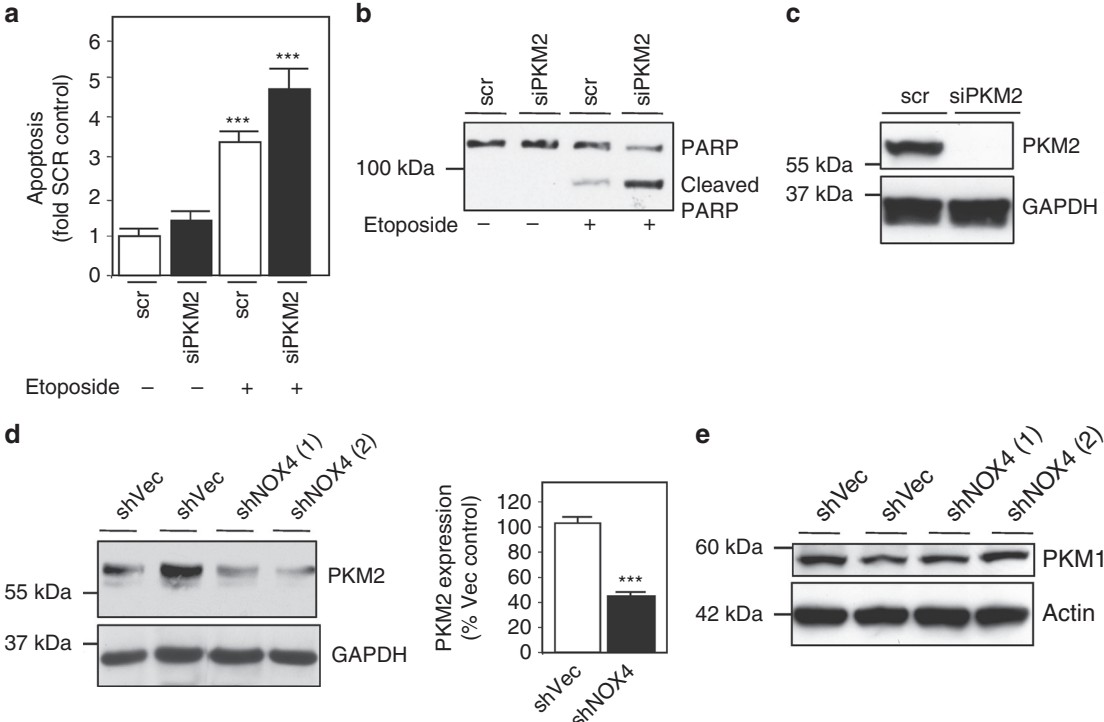

**Fig. 5** PKM2 is a novel mediator of RCC drug resistance. **a** RCC 786-O cells were transfected with scrambled RNA (scr) or small inhibitory RNA (siRNA) specific for PKM2 (siPKM2) using Amaxa. After 48 h, the transfected cells were incubated (+) with or without (−) etoposide (20 μM) for 14 h. Apoptosis was analyzed by Annexin V staining and flow cytometry. The results are presented from at least three independent experiments and expressed as the means where ±S.E.M. ***$p < 0.001$ compared to non-treated scr or siPKM2 control (scr). **b** In parallel to **a**, cell lysates were prepared and PARP cleavage was examined by western blot analysis. **c** Successful PKM2 downregulation was verified by western blot analysis. GAPDH was used as a loading control. **d** (Left panel) PKM2 expression was examined by western blot in shVector or shNOX4 RCC 786-O stable cell lines. GAPDH was used as a loading control. (Right panel) Quantitation of **d** using one-way ANOVA with Tukey's post hoc test where ±S.E.M. ***$p < 0.001$ is compared to shVector control. **e** PKM1 expression was examined in parallel to **d**. Actin was used as a loading control

(Fig. 6h). Together, this suggests NOX4 regulates PKM2 expression through a PCAF-dependent, acetylation- and lysosomal-mediated mechanism and that PKM2 is the critical NOX4 downstream target involved in RCC drug resistance.

**Pre-clinical characterization of NOX4 in drug-resistant RCC.** We injected 786-O cells stably transfected with shVector into the left flank or shNOX4 [(shNOX4[1] or shNOX4[2]] into the right flank of nude mice. After the tumors reached sizable levels (~5 weeks after initial injection), the mice were given one injection of etoposide for four consecutive days. Tumor growth was monitored using Calipers before the first injection of etoposide (start) and every day for 8 days total, marking the end of the experiment. Fold growth was analyzed by comparing measurements at the start of etoposide and at the end of the 8-day period, where the start was normalized to 1. We find VHL-deficient cells stably transfected with shVector continue to grow during etoposide delivery, whereas the shNOX4 RCC cells showed reduced tumor volume compared to the start of etoposide injection (Fig. 7a). To determine whether this may be due to changes in cell growth or increased apoptosis, we examined the effects of NOX4 and PKM2 knockdown on cell growth using thymidine incorporation[37]. We find that stable or transient knockdown of NOX4 or PKM2, respectively, did not reduce cell growth (Supplementary Fig. 10a, b, respectively). On the other hand, shNOX4 tumors plus etoposide show enhanced PARP and Caspase-3 cleavage compared to the shVector controls and lower PKM2 expression (Fig. 7b upper and lower panels, Fig. 7c). Actin and GAPDH were probed as loading controls respectively. Together suggesting the

tumor regression seen in shNOX4 tumors was due to increased cell apoptosis.

We next examined the expression of NOX4, PKM1, and PKM2 in isolated mitochondria from human RCC histologically classified as clear cell and normal uninvolved tissue from the same individual. We find NOX4 expression is increased in the mitochondria of RCC tissue compared to normal control (Fig. 8a, b). Importantly, we find ATP efficiently inhibited NOX activity in isolated mitochondria from tumors (Fig. 8c). Figure 8d shows marked increase in PKM2 expression in RCC tumors compared to normal, whereas PKM1 expression appeared unchanged.

To determine whether NOX4 drives drug resistance in human RCC, we freshly isolated RCC cells from patients undergoing partial or full nephrectomy as described[38]. The cells were cultured ex vivo and subsequently transfected transiently siRNA NOX4 (siNOX4) or scr control followed by exposure (+) or not (−) with doxorubicin. We find NOX4 silencing sensitizes human ex vivo RCC cells to drug-induced cell death compared to scrambled control (Fig. 9a–c upper panels). A representative picture of each human cell line is presented (Fig. 9a–c lower panels). Similar to the human tissue, we find PKM2 expression is higher in ex vivo patient cells compared to normal renal proximal tubular epithelial cells (Fig. 9d), whereas PKM1 expression was unchanged (Fig. 9e). Notably, NOX4 expression was higher in the total and mitochondrial fractions (Fig. 9f, g, respectively).

## Discussion
We provide the first evidence that the NADPH oxidase isoform, NOX4, is a novel energetic sensor within the mitochondria, which

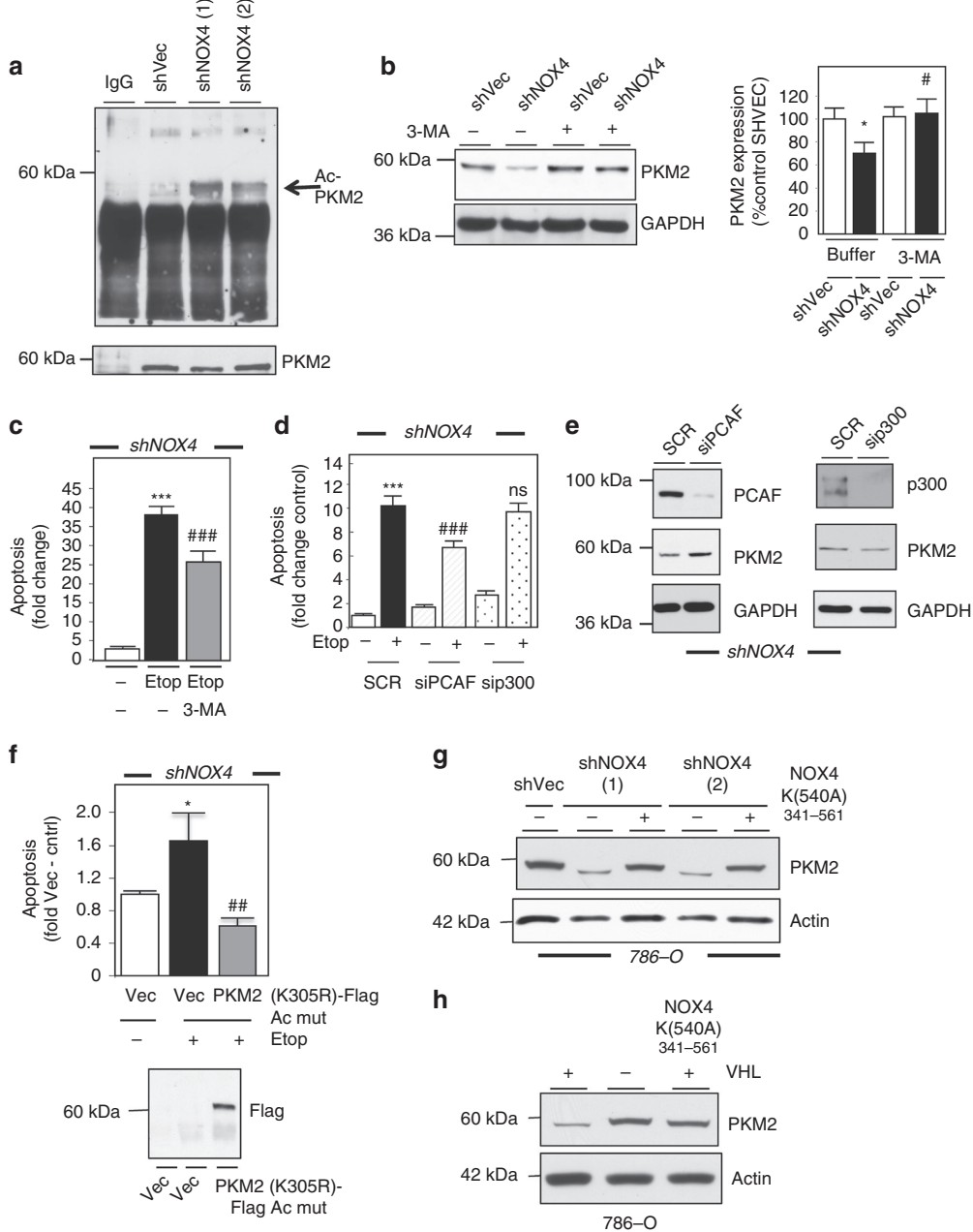

**Fig. 6** NOX4 regulates PKM2 stability through PCAF-dependent acetylation and lysosomal mechanisms. **a** PKM2 was immunoprecipitated using PKM2 antibodies in shVec or shNOX4 786-O cells and acetylation assessed using acetylation antibodies (noted by the arrow). PKM2 pull down was validated (lower panel). **b** (Left panel) PKM2 expression in shVec and shNOX4 786-O cells treated (+) or not (−) with 5 mM 3-MA for 24 h. (Right panel) Quantitation of **b** where ±S.E.M. *$p < 0.05$ is compared to shVec without (−) 3-MA and #$p < 0.05$ compared to shNOX4 without (−) 3-MA. **c** Apoptosis was assessed in shNOX4 786-O cells pre-treated (+) or not (−) with 3-MA followed by exposure (12–14 h) with etoposide or buffer (−) where ±S.E.M. ***$p < 0.001$ is compared to non-treated shNOX4 (−) and ###$p < 0.001$ compared to shNOX4 etoposide. **d** shNOX4 786-O cells were transfected with scrambled or siRNA specific for PCAF or p300. After 24 h, the cells were treated (+) or not (−) with 20 µM etoposide (12–14 h). The results are expressed as the means where ±S.E.M. ***$p < 0.001$ is compared to shNOX4 (scr) control without Etop and ###$p < 0.001$ compared to shNOX4 (scr) + Etop. NS represents non-significance. **e** In parallel to **d**, PKM2 expression was assessed and knockdown of PCAF and p300 were validated. GAPDH was used as a loading control. **f** (Upper panel) shNOX4 786-O cells were transfected with PKM2 acetylated mutant (K305R), using lipofectamine, for 24 h and subsequently exposed to 20 µM etoposide. Apoptosis was evaluated. The results are expressed as the means where ±S.E.M. *$p < 0.05$ is compared to non-treated Vector (Vec) control and ##$p < 0.01$ compared to shNOX4 cells transfected with Vec control + Etoposide. (Lower panel) Overexpression of PKM2 (K305R)-Flag acetylation mutant was verified by immunoprecipitation of TCL using Flag (anti-mouse) and western blot analysis using Flag (anti-Rabbit) antibody. **g** PKM2 expression was assessed by western blot analysis in 786-O cells stably silenced by NOX4 or shVector control with transfection of the ATP mutant NOX4 (K540A). **h** PKM2 expression was examined in 786-O with or without VHL, transfected (+) or not (−) with ATP mutant NOX4 (K540A). Actin was used as loading control. Statistics are expressed as the means using one-way ANOVA with Tukey's post hoc test

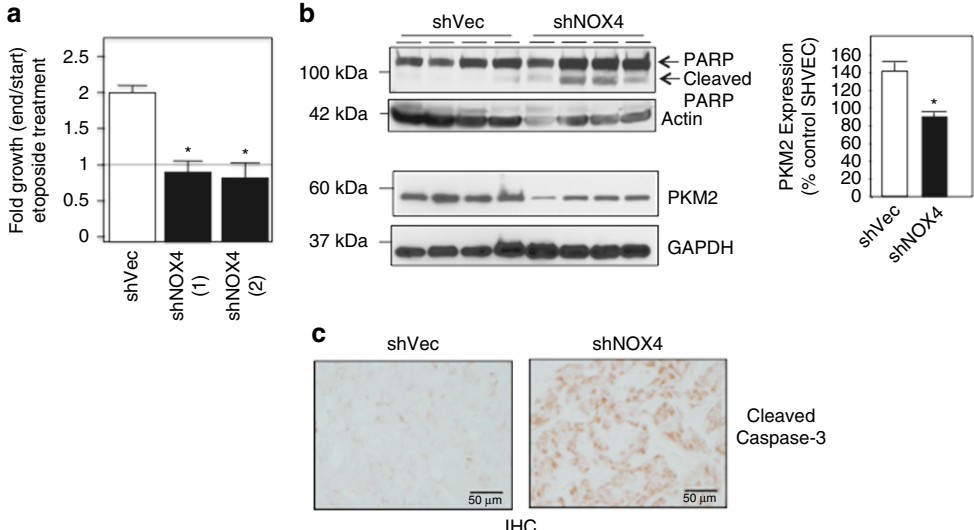

**Fig. 7** NOX4 inhibition sensitizes RCC cells to drug-induced cell death in vivo xenograft animal models. **a** Xenograft nude mouse model. shVector or shNOX4 cells were injected into the right and left flanks, respectively, of the same nude mouse. After 5 weeks of growth, etoposide was delivered IP for four consecutive days. Tumor volume was measured using calipers prior to the first etoposide injection and 4 days following the last etoposide injection. The results are from 10-independent mice for each independent shNOX4 clone[1, 2] and fold growth (end measurement/start measurement) is expressed as the means using one-way ANOVA with Tukey's post hoc test where ±S.E.M. *$p < 0.05$ is compared to shVector. **b** Tumors, from shVector and shNOX4 injections were removed from nude mice and lysates prepared. Total and cleaved PARP (upper panel) or PKM2 expression (lower panel) was examined by western blot analysis. Actin and GAPDH were probed as loading controls, respectively. Quantitation of **b** (left lower panel) expressed as the means using one-way ANOVA with Tukey's post hoc test where ±S.E.M. *$p < 0.05$ is compared to shVector. **c** Cleaved caspase-3 was examined by immunohistochemical (IHC) analysis in tumors isolated from shVector and shNOX4 xenograft mice exposed to etoposide. Microscopy (×10) of the representative images. Scale bar: 50 μM

serves as a checkpoint to couple mitochondrial energy metabolism to drug resistance in cancer cells. We suggest that during normal respiration, OXPHOS-driven ATP production in the mitochondria allosterically binds NOX4 through the Walker A, ATP-binding domain keeping NOX4-derived ROS low. Cellular events, such as cancer, which switch ATP production to aerobic glycolysis in the cytosol, thereby reducing ATP levels in the mitochondria relieves the breaks leading to NOX4 activation. Moreover, our data show that metabolic reprogramming-mediated activation of NOX4 inhibits PCAF-dependent acetylation- and lysosomal-mediated degradation of the oncogenic M2-PK (Fig. 9h).

NADPH oxidases of the NOX family are membrane-bound enzymes, which use NADPH as a substrate to transfer electrons across membranes to molecular oxygen to generate superoxide, which is subsequently dismutated to hydrogen peroxide. Our laboratory was the first to show, and observed by others, that NOX4 localizes to the mitochondria in renal, endothelial, and other various cell types[8, 16, 39–43]. Herein we find that NOX4 localizes to the inner mitochondrial membrane or to the inside of the outer mitochondrial membrane and is overexpressed in renal carcinoma cells in vitro and in human RCC ex vivo. Alternatively spliced variants of NOX4 have been detected in intracellular organelles such as the nucleus and endoplasmic reticulum of different cell types;[12] however, we show that mitoTempol (a mitochondrial superoxide scavenger) is necessary and sufficient to sensitize VHL-deficient RCC to drug-induced cell death. A role for NOX4 upstream or downstream of the metabolic switch remains unclear. NOX4-derived ROS has been shown to alter ETC activity in HUVEC cells. Our data demonstrate a novel mechanism by which NOX4 is activated (loss of ATP levels within the mitochondria) during aerobic glycolysis. Together, this suggests a novel NOX4-driven perpetual loop, but likely reversible, exists between aerobic glycolysis and mitochondrial

dysfunction. The regulation of NOX4 activity is not well understood. Unlike other NOX catalytic subunits, NOX4 does not require cytosolic proteins for activation[44]. p22phox, a small membrane bound protein is non-dispensable for activation of a subset of NOX catalytic subunits[12, 44]. However, an obligatory role for p22phox-dependent NOX4 activation has not been well established. We used the MitoProt II 1.0a4 prediction program to determine the likelihood of p22phox to be targeted to the mitochondria[8]. We find, unlike NOX4, which displays a 97% probability of being targeted to the mitochondria, p22phox showed a 0.02% chance of mitochondrial localization (Supplementary Fig. 11a). We fractionated 786-O and A498 into mitochondrial and cytosolic fractions and examined p22phox expression by western blot analysis. We find p22phox is predominantly found in the post-nuclear cytosolic fraction where predominant intracellular organelle or plasma membranes would fractionate (Supplementary Fig. 11b). Together, our data suggest NOX4 within the mitochondrial compartment is regulated by p22phox-independent mechanisms; although we do not exclude a role for p22phox-mediated NOX4 activation in other subcellular compartments.

This study defines a novel mechanism by which mitochondrial-localized NOX4 is allosterically regulated through adenosine triphosphate (ATP) levels. This conclusion is supported by our data showing NOX4 harbors a Walker ATP-binding motif, binds ATP directly, and is specifically negatively regulated by ATP, but not adenosine diphosphate (ADP) or guanosine triphosphate (GTP). These studies demonstrate qualitative, but specific, association of NOX4 with ATP. Outside the scope of this work, structure/function studies are needed to carefully quantitate stoichiometric analysis between NOX4 and ATP using full-length protein. The Walker-A binding motif is strategically located in the C terminus near the NADPH-binding site, a co-factor necessary for NOX activity. Further studies are needed to determine whether

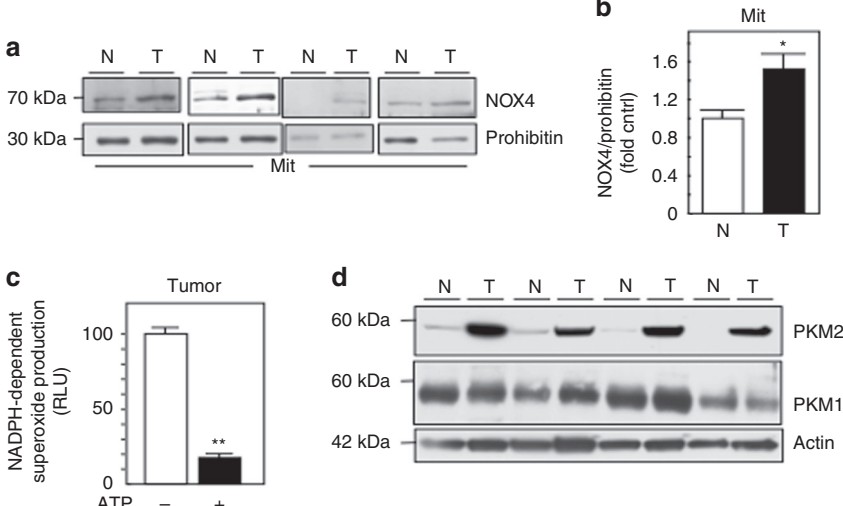

**Fig. 8** Characterization of NOX4 and PKM2 in human RCC tumors and adjacent tissue. **a** Mitochondrial fractions were prepared from human tumors (T) or uninvolved adjacent tissue (N). NOX4 expression was examined by western blot analysis. Prohibitin was probed as a mitochondrial marker and loading control. **b** Quantitation of NOX4 distribution in the mitochondrial fraction from **a**. The results are expressed as the means using one-way ANOVA with Tukey's post hoc test where ± S.E.M. *$p < 0.05$ compared to normal (N). **c** Mitochondria fractions were prepared from RCC tumors and NADPH-dependent superoxide generation was examined in the presence (+) or absence (−) of ATP. The results are from eight tumors and are expressed as the means using one-way ANOVA with Tukey's post hoc test where ±S.E.M. **$p < 0.01$ is compared to without (−) ATP. **d** PKM2 and PKM1 expression was examined by western blot analysis in lysates prepared from human tumors (T) or uninvolved adjacent tissue (N) from the same patient. Actin as loading control

NADPH binding to NOX4 blocks ATP-dependent binding and inhibition of NOX4 activity or whether ATP blocks NADPH-dependent activation of NOX4.

We additionally show that NOX4 contributes to the generation of ROS within the mitochondrial compartment and find that loss of the VHL protein results in lower ATP levels concomitant with enhanced NOX activity. On the other hand, re-introduction of VHL enhances mitochondrial ATP levels, concomitant with a reduction of NOX activity within the mitochondrial compartment. This observation was also supported by an independent approach where substituting galactose as the source of sugar in lieu of glucose enhances endogenous OXPHOS-driven ATP levels concomitant with a reduction of NOX activity within the mitochondrial compartment.

RCC cells are characteristically resistant to drug-induced cell death. The balance between production of ROS and their neutralization play a critical role in cell survival. Although enhanced ROS generation has been steadily suggested to induce cell death, we and others, find paradoxically that NOX4-derived ROS mediate cell survival[15–18]. This is supported by our finding that overexpression of an ATP mutant of NOX4, blocked drug-induced cell death in RCC cells re-expressing VHL.

Herein, we show that metabolic reprogramming-driven NOX4 activation mediates RCC drug resistance through PKM2. The *PKM* gene encodes for two isoenzymes, M1- and M2-PK where individual expression arises through alternative splicing[45]. Studies show PKM1 and PKM2 expression can co-exist in the same tissue or cancer cell type[32, 46–51]. PKM2 is often expressed at higher levels in cancer cells than PKM1; however, PKM1 has higher activity than PKM2[52, 53]. Additionally, PKM1 is constitutively active, whereas PKM2 can switch between active tetrameric and inactive dimeric forms[54]. Post-translational modification of PKM2 favors the inactive dimeric form, which is capable of rerouting glucose-derived metabolites toward anabolic synthetic pathways to support rapid cell growth and survival[55, 56]. Hence, directional steering of endogenous metabolites, mediated by PKM isoenzymes are complex and are governed by the stoichiometric

expression of PKM1 and PKM2, the modulation of their respective activities, and can also fluctuate contingent on the underlying cellular environment.

We find that NOX4 inhibits PKM2 acetylation-mediated lysosomal-dependent degradation as PKM2 acetylation is enhanced in shNOX4 clones and is stabilized in the presence of the lysosomal inhibitor, 3-MA. Furthermore, 3-MA reduces drug-induced cell death and PARP cleavage in shNOX4 RCC cells. PKM2 has multiple acetylation sites, which lead to distinct biological outcomes. PKM2 acetylation on K433 by p300 acetyl-transferase drives PKM2 to the nucleus whereas PKM2 acetylation on K305 by PCAF targets it to the lysosome for subsequent degradation[34, 57]. We show siRNA knockdown of PCAF, but not p300, sensitizes RCC cells to drug-induced cell death. Importantly, we suggest that acetylation- and lysosomal-driven reduction of PKM2 expression levels in NOX4 knockdown cells are necessary and sufficient to sensitize RCC cells to drug-induced cell death. To our knowledge, this is the first demonstration linking NOX4 and PKM2. The mechanism(s) by which NOX4 inhibits PCAF-dependent acetylation of K305 on PKM2 requires further investigation.

Importantly, we find RCC cells express both the M1- and M2-PK isoenzymes, but only PKM2, not PKM1, expression is regulated by NOX4. We hypothesize that in NOX4-silenced cells where PKM2 levels are reduced, PKM1 expression is sufficient to drive glucose-derived metabolites toward pyruvate generation, thus reducing anabolic-supported cell growth and survival. In support of this, our data show that silencing of NOX4 or PKM2 in RCC cells does not change nucleotide incorporation.

The clinical importance of our data is supported by our findings that NOX4 expression is enhanced in the mitochondrial compartment in 80%+ of human RCC tumors that correlates with upregulation of PKM2 compared to normal adjacent tissue from the same individual. Moreover, our preclinical animal models suggest stable NOX4 silencing in RCC cells enhances drug-induced dell death compared to shVector cells in nude mice.

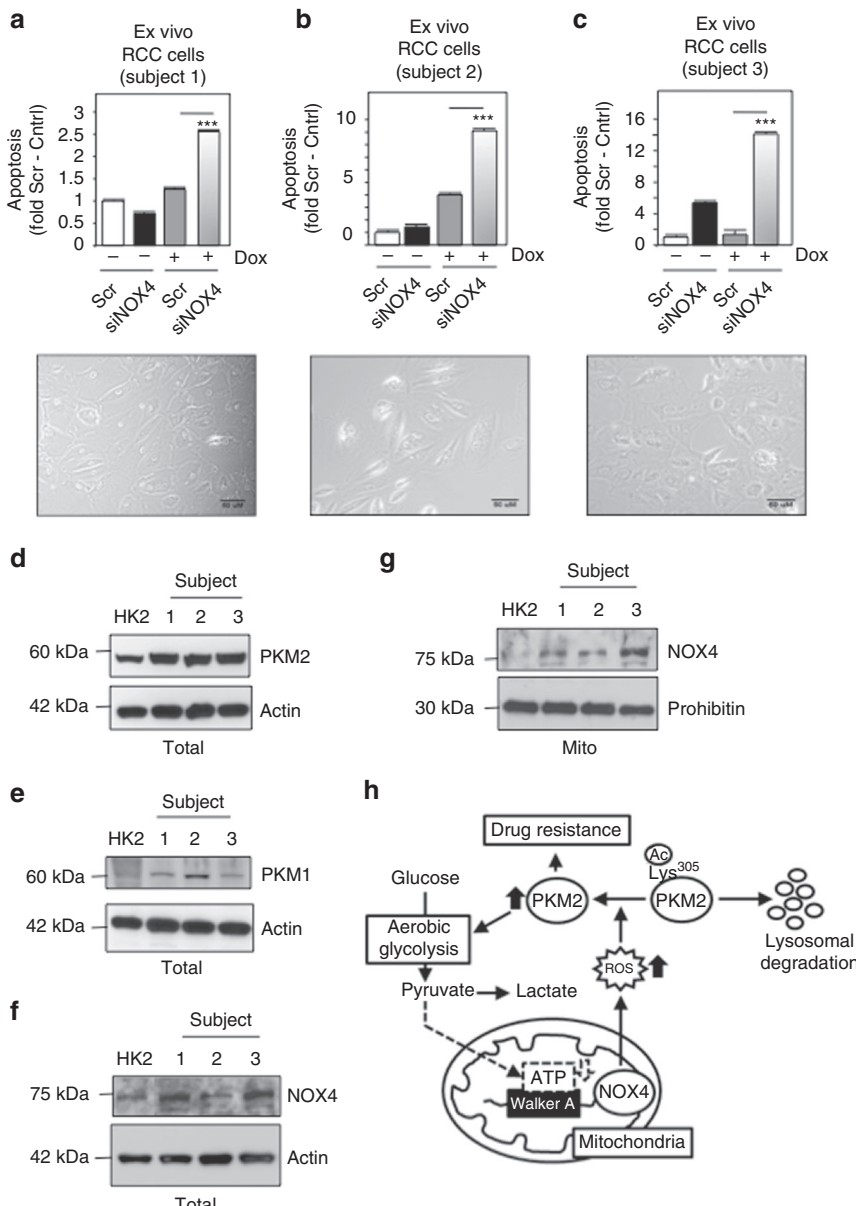

**Fig. 9** NOX4 inhibition sensitizes ex vivo human RCC cells to drug-induced cell death. **a–c** (Upper panel) RCC cells were isolated from human tumor tissue and established in culture. Ex vivo cells were transfected with siRNA to NOX4 (siNOX4) or scrambled control (scr). Subsequently, the cells were treated with doxorubicin and apoptosis was analyzed by Annexin V and flow cytometry. (Lower panel) microscopy of the representative human established ex vivo RCC cell lines from **a** to **c**, Scale bar: 50 μM. The results are presented from two independent experiments and expressed as the means using one-way ANOVA with Tukey's post hoc test where ± S.E.M. ***$p < 0.001$ compared to scr control + doxorubicin. Total cell lysates (Total) or mitochondrial (Mito) fractions were prepared from normal HK2 cells or ex vivo human RCC cell lines and **d** PKM2, **e** PKM1, and **f, g** NOX4 expression was assessed by western blot analysis. **h** Schematic model of NOX4's role coupling mitochondrial energetics to cancer drug-resistance through PKM2

Together, we provide the first evidence that NOX4 functions as a mitochondrial energetic sensor and serves as a novel metabolic checkpoint, coupling the metabolic switch to cancer cell survival. We posit that targeted small molecules which mimic ATP binding to the NOX4 Walker A-binding site would be novel therapeutic approaches to reduce drug-resistance in renal cancer and likely other cancer cell types in which NOX4 drives drug resistance.

## Methods

**Chemicals and reagents**. Rotenone, ATP, ADP, GTP, etoposide (E1383), BSA, 3-Methyladenine (3-MA), lucigenin, proteinase K, Doxorubicin, Mitotempol, and IPTG (Sigma-Aldrich, St. Louis, MO), NADPH (Roche Diagnostics, Indianapolis, IN). Mitotracker Deep Red and MitoSOX (Molecular Probes).

**Cell culture**. Normal human renal epithelial cells, HK2 (American Type Culture Collection, ATCC) were grown in Dulbecco's modified Eagle's medium (DMEM)-F12 (Gibco Life Technologies, Grand Island, NY) supplemented with 10% FBS, 2 mM L-glutamine and 1× antibiotic. VHL-deficient renal carcinoma 786-O and A498 cell lines (ATCC) were maintained in DMEM HG, 10% FBS, 2 mM L-glutamine, and 1× antibiotic as described[7], and 786-O (VHL) with supplemented puromycin (350 ng/ml) as described[24]. All cells were maintained at 37 °C with 5% $CO_2$. In some experiments, RCC 786-O or A498 cells were switched to glucose-free media supplemented with galactose (10 mM) supplemented with 4 mM L-glutamine for 20–24 h. Isolation of primary RCC cells was described[38]. Briefly, renal tumor samples and normal corresponding tissue from patients undergoing nephrectomy were obtained from the Department of Urology at the UTHSCSA according to an approved non-human non-identifiable Institutional Review Board IRB protocol #HSC 20070777N. For samples prepared for cell culture, unwanted renal tumor tissue was placed in a sterile 50 ml conical tube containing ice-cold

culture medium and transferred to the laboratory within 20 min of collection. Subsequently, the fibrous capsule and medulla were dissected out and the tissue was cut into small pieces on a Petri-dish and rinsed. The tissue was minced further, washed, and treated with collagenase solution (1 mg/ml) for 20 min at 37 °C. The digested tissue was passed over 100, 70, and 40 μM sieves to remove tubular fragments and glomeruli. This fraction was washed 3× and the cell pellet was cultured in medium. After the cell culture achieved 80% confluence, experiments were conducted as outlined.

Mycoplasma contamination was assessed in all cell cultures on a periotic basis. No contaminated cells were used in the experiments outlined in this study.

**Cell transfection**. For transient knockdown experiments, the indicated cells were transfected by nucleofection using a nucleofector (Amaxa Biosystems, Allendale, NJ) according to the manufacturer's protocol. For optimal transfection, RCC 786-O cells, kit V (VCA-1003) was used where for A498 cells, kit L (VCA-1005) was used. Briefly, cells ($2.0 \times 10^6$) were pulsed with 3 μM smart pool siRNA against NOX4, PKM2, VHL FLAG, PCAF, p300 or scrambled (scr) control (Dharmacon) and plated in the respective growth media. Cells were harvested typically 24–72 h posttransfection for the specific assay to be tested. In overexpression experiments, transient transfection was carried out using Lipofectamine 2000 reagent (Invitrogen, Grand Island, NY).

**Generation of shNOX4 stable single-cell knockdown clones**. MISSION Lentiviral transduction particles for vector control or human shNOX4 (Sigma-Aldrich) were used to generate stable knockdown clones in RCC 786-O cells. Mission Lentiviral transduction particles (SHCLNV product#; NM_016931 geneID; Human): (Sigma Aldrich): NM_016931 TRCN0000046089 (Sequence: CCGGGCT GTATATTGATGGTCCTTTCTCGAGAAAGGACCATCAATATACAGCTTT TTG) and TRCN0000046090 (Sequence:CCGGCCCTCAACTTCTCAGTGAATT CTCGAGAATTCACTGAGAAGTTGAGGGTTTTTG). 786-O cells were infected with Lentiviral shNOX4 or shVector control as per the manufacturer's instructions. Twenty-four hours post transfection, single cells were selected and passed into 96-well plates in puromycin (950 ng/ml) antibiotic for selection and maintained in puromycin (350 ng/ml).

**SPR analysis**. WTNOX4 and mutNOX4 binding affinity to ATP using surface plasmon resonance-based technology (Pioneer SensiQ, Sinsiq Technologies, performed by Affina Biotechnologies, Elmsford, NY). NOX4 (both WT and mutant) were immobilized on a HisCap chip (SensiQ Technologies). For immobilization, the HisCap chip was charged with Ni and activated with 0.05 M NHS/0.2 M EDC for 3 min. The protein was injected at 50 μg/ml on the activated surface to achieve the maximal loading of 1300RU for both wild-type and mutant proteins after which the surface was deactivated with 1 M ethanolamine. Immobilization was performed in HBS-P at 25 °C, while binding was tested at 4 °C in the binding buffer as above. Samples were injected in a FastStep injection mode (SensiQ Tehcnologies) resulting in a sequential injection of 1.56, 3.13, 6.25, 12.5, 25, 50, 100 μM ATP followed by a single dissociation step. The reference surface was activated and deactivated in the same manner and did not contain any protein.

**ATP filter binding assay**. ATP binding to recombinant (r) proteins, rNOX4[341–561] and rNOX4[341–561MUT] were analyzed by the filter-binding assay[21]. Briefly, indicated rNOX4 proteins (1 μg) were incubated with $\alpha^{32}$P-ATP (0.125–1 μCi) for 30 min on ice in 50 μl of binding buffer (50 mM tricine, 0.5 mM magnesium acetate, 0.3 mM EDTA, 7 mM DTT, 20% glycerol, and 0.007% TritonX-100). Samples were then spotted on 0.45 μM nitrocellulose filter, washed three times (10 min each) in wash buffer (50 mM Tricine, 0.5 mM magnesium acetate, 0.3 mM EDTA, 5 mM DTT, 17% glycerol, 10 mM ammonium sulfate, and 0.005% TritonX-100) to best rid of unincorporated $\alpha^{32}$P-ATP. The nitrocellulose filters were put into scintillation vials and the radioactivity remaining on the filters was counted in a liquid scintillation counter. The results are expressed as DPM to background where the minor background noted is contributed by noise to signal $\alpha^{32}$P-ATP stuck to the membrane.

**Measurement of oxygen consumption rate**. Basal oxygen consumption rate (OCR) was measured in live cells using the Seahorse XF24 extracellular flux analyzer (Seahorse Bioscience Inc., N. Billerica, MA). Cells (25,000 cells/well in 250 μl culture media) were seeded in XF24 microplate. Next day, 200 μl of culture medium was removed from each well and the cells were washed in 1 ml of assay medium. After washing, 550 μl of assay medium was placed in each well and the microplate was incubated for 1 h at 37 °C in a non-CO₂ incubator and then analyzed by the XF24 extracellular flux analyzer. The OCR was measured using the following protocol command (mix 3 min, time delay 2 min, and measure 3 min) in the XF24 analyzer.

**Measurement of ATP**. ATP was measured using the luminescence ATP detection assay system (PerkinElmer, Boston, MA) as per the manufacturer's instructions. Briefly, cells (10,000/well in 100 μl volume) were seeded in a 96-well white microplate. Next day, 50 μl of cell lysis solution was added to each well and the

plate rotated for 5 min using an orbital shaker at 700 rpm to lyse cells and stabilize ATP. Note: for experiments using rotenone, specified cells were treated with rotenone 1 μM for 30 min before all cells received cell lysis solution. Then 50 μl of substrate solution was added to all the wells and the plate rotated for 5 min using the orbital shaker. The plate was kept in dark for 10 min and luminescence was measured using microplate reader. The emitted light is proportional to the ATP concentration.

**Determination of apoptosis**. RCC 786-O or A498 cells (transfected or not) were typically exposed to etoposide (20 or 100 μM, respectively) or cultured or freshly isolated RCC cells to doxorubicin (1 μg/ml) for 12–14 h. Cells were washed, trypsin, and ~300,000 cells were stained with Annexin and PI followed by apoptosis assessment. Early (Annexin V positive; PI negative) and late apoptosis (Annexin V positive; PI positive) were detected using the Annexin V-FITC apoptosis detection kit (EMD Millipore, San Diego, CA). Apoptotic cells were analyzed by flow cytometry (UTHSCSA, Flow Cytometry Core). For pre-treatments, 3-MA (5 mM) was pre-treated for 24 h, Mitotempol (500 nM) was pretreated for 6 h, or galactose for 20 h, followed by addition of etoposide for 12–14 h. For transient over-expression of NOX4mut or PKM2mut, after 24 h cells were exposed to etoposide for 12–14 h. For transient knockdown of NOX4 (A498) or PKM2, the cells were transfected with nucleofection and 24 h, etoposide was added for 12–14 h. Ex vivo cells were transfected with siNOX4 using nucleofection (kit V) 24 h followed by 12–14 h exposure to doxorubicin (1 μg/ml).

**Measurement of DNA synthesis**. DNA synthesis was measured as incorporation of [3H]-thymidine into trichloroacetic acid (TCA)-insoluble material[37]. Briefly, shVector or shNOX4 cells were plated into 12-well dishes and the next day washed with PBS and pulsed with 1 mCi/ml [3H]-thymidine for 4 h. The medium was removed and the cells were washed twice with ice-cold 5% TCA to remove unincorporated [3H]-thymidine. Cells were solubilized by adding 0.75 ml of 0.25 N NaOH and 0.1% SDS. A 0.5 ml volume of this cell lysate was neutralized and counted in a scintillation counter.

**Mitochondrial/cytoplasmic fractionation**. Where indicated, Mitochondrial and cytosolic fractionation was performed using the mitochondria isolation kit (Pierce Biotechnology, Rockford, IL) with minor changes where the post-debris fraction was subjected to a 3000 × g centrifuging for 15 min to further reduce of lysosomal and peroxisomal contaminants. In experiments where biological activity was assessed, crude mitochondria fractions were prepared as outlined in ref. [8]. For Proteinase K Digestion experiments, Percoll gradient-purified mitochondria were prepared as described[8]. Equal concentrations of Percoll gradient-purified mitochondria were exposed to 0.1 nM of Proteinase K digestion for T0, T5, T10, and T15 min at RT. The reaction was stopped using 2× Laemmli sample buffer.

**Western blot analysis and immunoprecipitation**. Cell and tissue lysates were prepared in ~250–500 μl radioimmune precipitation assay (RIPA-) buffer (10 mM Tris, pH 8, 1 mM EDTA, 150 mM NaCl, and 1% NP 40) using a Dounce homogenizer. Cell and tissue lysates were rotated on a rotator for 2 h and then centrifuged at 12,000 × g for 15 min at 4 °C. After centrifugation, the protein content in the supernatant was measured using the Bradford reagent (Bio-Rad, Hercules, CA). For western blotting, ~30–50 μg of lysate were electrophoresed on a SDS-PAGE and the separated proteins from gel were electro-transferred on to nitrocellulose membrane. The membrane was subjected to a 60 min blocking step according to the primary antibody manufacturer's suggestion. For western blot or IHC, primary antibodies were used as per the manufacturer's recommendation: prohibitin 1 μg/ml (Abcam ab28172), PARP 1:1000 (Cell Signaling 9542), Cleaved Caspase-3 1:1000 (Cell Signaling 9661), 6xHIS 1:1000 (Cell Signaling 2365), VDAC 1:000 (Cell Signaling 4866), PCAF 1:1000 (Cell Signaling 3378), p300 1:500 (Millipore NA46), GAPDH 1:250 (Santa Cruz sc25778), Actin 1:2000 (Sigma A2228), Flag (Sigma F3165 1:100,F7425 1:300), Tubulin 1:4000 (Sigma T9026), NOX4 1:500 (Novus NB110-58849), our NOX4 1:1000[8], p22phox 1:500 (Santa Cruz), LDH 1 μg/ml (Abcam ab55433), COX 2 μg/ml (Mito-Science MS414), M1-PK 1:400 (Novus NBP2-14833), GAPDH 1:20,000 (Sigma), PKM2 1:1000 (Cell Signaling 4053S), Acetylated Lysine 1:1000 (Cell Signaling 9441). The indicated primary antibodies were probed at 4 °C overnight. The membrane was washed and then incubated for 1 h with appropriate horseradish peroxidase (HRP) conjugated secondary anti-bodies anti-mouse (NA931) or anti-rabbit (NA934-GE) were used. The membrane was washed and developed to visualize protein bands using enhanced chemiluminescence (ECL) reagent (GE Healthcare UK Limited, Buckinghamshire, UK) or with SuperSignal West Pico Luminol/Enhancer solution (Thermo Scientific, Rockford, IL). Representative full gels for important experiments are presented in supplemental information (SI). For immunoprecipitation of PKM2 (PKM2 IP), indicated cells were lysed in buffer containing 0.3% CHAPS and centrifuged at 10,000 × g for 10 min at 4 °C. Protein was determined in the cleared supernatant using the Bio-Rad protein assay reagent. For immunoprecipitation experiments: PKM2, equal amounts of protein (1000 μg) were incubated with 3 μG of Rabbit anti-PKM-2 antibody (Cell Signaling Technology) and Protein A-Sepharose beads (GE Healthcare Bio-Sciences) and rotated overnight at 4 °C. The beads were washed three times with CHAPS lysis buffer and suspended in 20 μl of

1× loading buffer. NOX4 IP: NOX4 (Novus-49), or IgG control was immuno-precipitated using protein-A sepharose beads from 250 µg of crude mitochondrial preparation each from MSHE buffer to maintain biological activity per condition as described in ref. [8]. The beads were washed and incubated (+) or not (−) with 5 mM ATP for 60 min on ice. Subsequently, NADPH oxidase activity was assessed using enhanced chemiluminescence on the beads. After the assay, the beads were recovered by centrifugation and boiled with 1× loading buffer and resolved on SDS-PAGE. Immunoblotting was performed using our NOX4 antibody[8]. Immu-noprecipitation of FLAG: (750 µg) of total cell lysate was used to immunopreci-pitate anti-FLAG as described. For all immunoprecipitation assays, secondary HRP-linked antibodies; anti-rabbit (NA9340-GE) or anti-mouse (NA9310-GE), were used to reduce IgG background detection.

**Site-directed mutagenesis, cloning, and purification of NOX4 recombinant proteins.** The Walker A, ATP-binding site in NOX4 was mutated from lysine to alanine (K540A) using site-directed mutagenesis (USB Biochemicals, Affymetrix, Santa Clara, CA). Primers were engineered as follows: top hNOX4K560A 5′-TATAACAGAGGAGCAACAGTTGGTGTTTTC-3′ and bottom hNOX4K560A 5′-GAAAACACCAACTGTTGCTCCTCTGTTATA-3′. Full-length hNOX4 pcDNA 3.1 was used as the template. Plasmid DNA was prepared from both WTNOX4 and mutNOX4 clones and correct sequences were confirmed by DNA sequencing. Subsequently, the human WT NOX4 and mutant NOX4 cDNA was cloned into pET-NusA-44 expression vector (Novagen, EMD Millipore, Germany) according to the manufacturer's instructions. In brief, the human NOX4 fragment (spanning from 1021 to 1681 bp) harboring the ATP-binding site was amplified by polymerase chain reaction (95 °C hot start 2 min; 95 °C 20 s, 58 °C 10 s, 70 °C 20 s for 25 cycles, 70 °C for 10 min) using specific primers (top: 5′-GACGACGACAA GATGCTACATTGTCCCAGTGTATCT-3′) (bottom: 5′-GAGGAGAAGCCCGG TTAGTTACTCAGTTTATGAAGAGT-3′). The amplified fragments were gel purified using the S.N.A.P. UV-Free gel purification kit (Invitrogen), treated with T4 DNA polymerase, and annealed with the pET-NusA-44 vector. The rNusA WT NOX4 and mutant NOX4 plasmids were then transformed into BL21 (DE3) expression *E. coli*. host. Soluble rNOX4 proteins were produced by growing the transformed cells in presence of 1 mM Isopropyl β-D-1-thiogalactopyranoside (IPTG) at room temperature for ~3 h. The recombinant proteins were purified using the BugBuster Ni-NTA purification kit (Novagen), dialyzed, and con-centrated using the Pierce concentrator (Pierce Biotechnology). Soluble purified proteins were analyzed by Coomassie blue staining.

**Immunofluorescence confocal microscopy (co-localization).** Cells were seeded and grown on four-well chamber slides and stained with 500 nM-1 µM Mito-Tracker Deep Red (MTR) 633 for 30 min. Cells were subsequently fixed with 4% paraformaldehyde for 15 min and permeabilized with 0.2% Triton X-100 for 5 min. The cells were blocked for 30 min and incubated with appropriate primary anti-bodies for 30 min. FITC-conjugated secondary antibodies were then applied and incubated for 30 min. The cells were washed three times with PBS, mounted with anti-fade reagent with DAPI, and visualized on Olympus FV-500 confocal laser scanning microscope. MTR 633 fluorescent intensity was determined at 650-nm excitation and 668-nm emission. All fields were taken at ×40.

**Detection of superoxide generation within the mitochondrial compartment.** Superoxide generation within the mitochondrial compartment was assessed in live cells using MitoSOX, a fluorogenic dye that is taken up by mitochondria where it is readily oxidized by superoxide, as previously described[8, 43]. Briefly, live cells were loaded with 1 µM MitoSOX Red in phenol-free DMEM for 10 min at 37 °C. Cells were washed with warm buffer. MitoSOX Red fluorescent intensity was determined at 515-nm excitation and 580-nm emission. All fields are ×10.

**Amplex red hydrogen peroxide assay.** Hydrogen peroxide production was measured fluoroscopically using Amplex Red Assay Kit (Invitrogen/Molecular Probes) in the indicated cell preparations as described[8]. The amount of hydrogen peroxide generated was determined by measuring the absorbance of the Amplex Red reagent at 565 nm.

**NADPH-dependent superoxide production measurements (NOX oxidase assay).** NADPH-dependent superoxide generation was measured in the presence or absence of 5 mM ATP, ADP, or GTP (pre-incubated for 60 min) in the indicated cell or tissue homogenates or crude mitochondrial fractions prepared in lysis buffer or 1× MSHE buffer using lucigenin-enhanced chemiluminescence method as previously described[7, 8, 10, 11, 13].

**Xenograft animal model.** Stable shNOX4 or vector control RCC 786-O cells were mixed with matrigel and injected into 6-week-old male athymic nude (BALB/c, nu/nu) mice as described[13]. Briefly, shVector was injected to the left flank and shNOX4 to the right flank of the same mouse. Two independent vectors and two independent shNOX4 clones were injected into 10 mice per group. To examine the effects of etoposide in RCC shVector and shNOX4 injected nude mice, we delivered four consecutive intraperitoneal (IP) injections of etoposide (Chemietek

ct-eph) (12 mg/kg/day) for a total of 4 days. The tumors were measured prior to etoposide delivery and at the end of the experiment (5 days after last etoposide injection). The tumors were measured again at the end of the experiment and the difference between tumor volume at the start and end were graphed normalized to 1, which indicates the start volume. Mice were then killed and the tumors excised and snap-frozen. All experiments using mice were approved by the University of Texas Health Science Center at San Antonio Institutional Animal Care and Use Committee.

**Statistical analysis**. Results are expressed as mean ± SE. Statistical significance was assessed by one-way analysis of variance (ANOVA) with Tukey's post hoc test where $p < 0.05$ was considered significant.

**Data availability**. The authors declare that all data supporting the findings of this study are available within the article and its Supplementary Information files and from the corresponding author upon reasonable request.

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

## Acknowledgements

We thank Drs. Y. Gorin, H.E. Abboud, and B.S. Masters for helpful discussions, Y. Gorin for creative model designs, and Dr. A. Valente for providing hNOX4 cDNA. We acknowledge UT Health- and the Cancer Therapy & Research Center (CTRC) P30-supported shared resources including FLOW cytometry and Optical Imaging. Our study was supported by grants from Veterans Administration Merit Award (K.B.) NIH DK033665 (K.B.), and NIH P30 CA054174 (K.B.).

## Author contributions

K.B. conceived the project, managed experimental design and data interpretation, and wrote the manuscript. K.S. primarily carried out the experiments herein and provided experimental details in materials and methods. W.E.F. handled all animal experiments. ATP in vitro binding assays were carried out by B.K.N. with the guidance of K.B. D.K. and R.R. provided freshly extracted unwanted human RCC and normal tissue.

## Additional information

**Competing interests:** The authors declare no competing financial interests.

