## [Peer Review File · Nature Communications]

Reviewers' comments:

Reviewer #1 (Remarks to the Author):

The manuscript entitled, 'NOX4 functions as a novel mitochondrial energetic sensor, coupling cancer metabolic reprogramming to drug-resistance' from Shanmugasundaram and colleagues describes a novel link between the NADPH oxidase, NOX4, mitochondrial energy metabolism and resistance to cell death. It is an interesting manuscript with good quality in vivo data, but variable quality in vitro studies. As such, although it is apparent that a novel mechanism exists, the validation of it is not of sufficient quality to be entirely convincing.

There is, in general, insufficient detail in the, figure legends, materials and methods or results text to be able to appropriately assess some of the data – for example, the number of times an experiment is repeated is unclear in most circumstances, as is the timing of treatments, concentration of inhibitors used, etc. Often appropriate controls are not included in the data. Specific examples include use of the shControl lines in Figure 6C and D, loading controls in Figure 4E right panel, loading controls for IP in Figure 6A, a mitochondrial marker such as mitotracker green whenever immunofluorescence data looking at mitochondria are shown and consistent use of both vector controls and both shNOX4 cell lines when showing data from this model. Data not shown is no longer suitable - all data referred to needs to be previously published, or shown in the manuscript.

As such the data is difficult to judge for rigour without appropriate details in materials and methods – specific examples include concentration of galactose used, which experiments were carried out using Lipofectamine 2000 and which using nucleofection, what the shRNA sequence(s) were that were used for shNOX4 knockdown, concentration of rotenone used in each experiment and for how long it was incubated with the cells before measurements were made and finally the NOX4 immunoprecipitation methods – which were not provided but referenced to a previous paper, which itself referenced previous papers for the method.

Throughout the manuscript, the Millipore Annexin V-FITC kit is used, which includes propidium iodide (PI) as an additional measure of cell death. It is not clear whether the population used to define 'apoptosis' in the analysis were just PI positive or both negative and positive. Data from this assay must be presented in a consistent manner – sometimes it is fold change compared to control, and sometimes it is '% apoptosis', defined as '%- Cntrl'. This can lead to a significant lack of clarity – for example, in Figure 4A, untreated shNOX4 (1) line has approx. 6% apoptosis, and an increase to around 22% is noted with etoposide – around a 4-fold change. In Figure 6C, what is presumably the same line (unclear) shows a fold change of 2.5 (unclear what '1' is set at) to over 35-fold. We are also not aware of the number of times this experiment was repeated, but the data appear highly inconsistent.

Specific Points

The figures are inaccurately referenced in the text – there is no Supplementary Figure 7 despite it being referred to in the results section text, and from around Supplementary Figure 3 onwards all of them are incorrectly referenced.

Please be clear whether a mitochondrial fraction (unhomogenized) or a homogenate was used in experiments – especially for ROS measurements (page 10).

Figure 1 – how are negative ATP incorporation measurements possible (panels C, D and E)

Figure 2 – panel A; images using mitotracker red need improvement, and more cells should be shown. Panel B; there is considerable data suggesting LDH (which isoform was blotted for – not clear) can be in the mitochondria, therefore a more appropriate marker should be used.

Figure 3 – panel A; a marker of mitochondria as well as their membrane potential must be used, as loss of staining could be through loss of mitochondrial weight, or change in mitochondrial membrane potential, which affects mitoSOX uptake. Panel E; expression of FLAG-pVHL should be confirmed in both cell lines used. GAPDH is an inappropriate loading control, due to its being a HIF1 target gene, and therefore its expression may change with pVHL re-expression.

Figure 4 – Both panel A and C show data using the 786-O cell line. In panel A, etoposide results in either no change in apoptosis, or a decrease in the 786-O stably expressing the empty vector. In panel C it appears to result in an over 3-fold increase. The empty vector appears to have a significant biological effect that alters the phenotype of the cells. Panel D; is the empty vector line not the most appropriate control to use here?

Figure 6 – panel A; it is not clear where the appropriate band is that represents acetylation of PKM2 – it would be clearer with more controls or a better system – previous reports have used FLAG-PKM2. Positive controls such as nicotinamide or trichostatin A would also be helpful. A loading control is also advisable.

Figure 7 – panel A; the assessment of fold growth in the xenograft four days after the last treatment provides the opportunity for the xenografts to start to proliferate again. As it is possible that the shVec and shNOX xenografts grow at different rates, this could significantly affect the result. It would be help clear this up if the untreated growth curves for each xenograft type could be shown. Panel B; was the assessment of PARP performed on the same tumours 4 days after the last treatment? If so, it is difficult to understand how this cleavage is still occurring so long after the last dose.

Supplementary Figure 2 – panel A; the data are also consistent with a protein that is on the inside of the outer mitochondrial membrane, not just something in the inner membrane, as suggested in the text. Panel E; the knockdown achieved in the A498 line appears not to have been successful, which calls into questions results from this model.

Reviewer #2 (Remarks to the Author):

Comments to the Authors

The study of Shanmugasundaram et al aimed to better understand the molecular mechanisms that couple glycolysis to cancer drug resistance. They identified the NADPH oxidase NOX4 as a novel mitochondrial energetic sensor whose ROS generating activity is directly regulated by the ATP level and impacts on the fat of the key metabolic regulator pyruvate kinase-M2 isoform.

This study is of importance. The experimental work is rigorous and well done. The authors present interesting and original data. There are however some concerns that should be addressed

Results

Measurement of NADPH-dependent superoxide production and H₂O₂ production of NOX4 (540A) mutant compared to NOX4 WT is lacking. The use of MitoSOX is not sufficient

Fig2: This figure highlights that NOX4 is mainly expressed in the mitochondrial compartment.

However, NOX4 is described to be expressed as several isoforms in different cell types and moreover, as membrane protein, it is synthesized in the endoplasmic reticulum. To confirm the enrichment of one NOX4 isoform in the mitochondrial fraction of these cells, the expression of the enzyme should be compared with the expression of NOX4 in endoplasmic reticulum but also in the nuclear fraction of these cells.

In the discussion the authors mentioned that, based on a prediction program, the p22phox, which is described as a functional partner of NOX4, is probably not expressed in the mitochondrial fraction of the RCC cells. This point is very important and interesting and needs to be illustrated by a western blot analysis of p22phox expression in different cellular fractions.

The authors should better define the cytosolic fraction. Is there the supernatant of the mitochondrial fraction? In this case the proteins are diluted and therefore NOX4 is less propitious to be detected in these conditions

Fig S2: In general there are mistakes concerning the Supplemental data in the manuscript. The corresponding numbers are wrong and the legends are missing.

S2C instead of S3A, S2D instead of S3B and S2E instead of S3C etc

This figure shows a significant reduction of NADPH-dependent superoxide production in isolated mitochondrial fractions silenced of NOX4 compared to controls. However, if we consider that H₂O₂ results from a dismutation of superoxide anions how authors explain that extinction of NOX4 suppresses 80 % of superoxide production and only 25% of corresponding H₂O₂ production?

Same question for the effect of ATP in figure 2C, 2D and 2E

How exogenous ATP impact on the NOX4 activity in the mitochondrial fraction? ATP needs to cross both outer and inner membranes of mitochondria before to interact with the carboxyl terminal domain of NOX4. Are there some channels or the mitochondrial fractions are made permeable when preparing? The Authors should specify.

ATP seems increase the immunoprecipitation of NOX4, is there correct ?

Fig 3: A western blot analysis of NOX4 expression should complete the diagrams to confirm that VHL does not impact on the level of NOX4 protein expression (This is only shown for 786-O cells in figure 2 but not for A498 cells).

VHL enhances OCR. It has been previously shown that NOX4 activity induces dysfunction of mitochondrial complex I in HUVECs (Koziel R et al, 2013 Biochem. J; 452:231-239). Therefore modulation of NOX4 activity seems to be important in the control of the mitochondrial function. The authors should discuss this point to the light of their data.

Fig 4C: How long the cells are treated with mitotempol ?

Fig 6 C and Fig 6D: Western blot analysis of PKM2 expressions are missing. In addition western-blot analysis of effect of siPCAF and sip300 on the expression of the respective proteins should be added.

Figure 7: The authors should complete the figure by analysing the expression of PKM2 in tumors by western-blot but also by IHC.

Figure 8: Once more the authors should complete the figure by western blot analysis of both NOX4 and PKM2 protein expressions in ex vivo RCC cells.

Effect of NOX4 extinction on PCAF expression should be evaluated

Reviewer #3: (Remarks to the Author):

"Nox4 is a novel sensor of mitochondrial ATP levels driving chemotherapy resistance in cancer cells: Role of PKM2" by K Shanmugasundaram and colleagues is a well-written manuscript laying out a series of elegant step-wise experiments that clearly define a role and mechanism of mitochondrial Nox4 regulation in primary and established renal cancer cells. It further demonstrates through in vitro and in vivo studies that this pathway is necessary and sufficient for resistance to chemotherapeutic drugs in these cells. This represents a key contribution that will be important for cancer therapeutics as well as ROS cell signaling.

The hypothesis and model are clearly described and the figures are clear and compelling. The text at times uses a wordy and repetitive style and could be shortened if space requires this. However, the flow is not negatively affected. An example is the final sentence of the first paragraph of the introduction.

I have no major concerns.

Minor comments:

1. introduction, paragraph 4 should read, "alternative splicing"

2. Introduction paragraph 5

a. first sentence should read, "and *has* the highest death rate..."

b. second sentence is missing an object. Suggest, "Despite surgery to remove the affected kidney, *many* [it is actually 30-40%] patients succumb to metastatic disease due to the lack of..."

3. In the same cell systems, we and others have isolated Nox4 expression and function from the ER membrane fraction as well. Did these fractionation studies include ER? If so, comment on the discrepancy.

Dr Jodie Maranchie.

Reviewer #4 (Remarks to the Author):

Given the short notice and the restricted time, my comments address only the SPR data reported by the authors. Indeed the latter claim that they have "evaluated the WTNOX4 and mutNOX4 binding affinity to ATP using Biocore surface plasmon resonance-based technology (Affina Biotechnologies)".

- It should be Biacore and not Biocore;
 - Authors state that "equal concentration of WT NOX4 (341-561) and mutant NOX4 (341-561 K540A)" were used: what was the concentration? Have the authors any ideas on surface coverage? What was the amount of immobilized proteins on the surface?
 - Was the reference cell response subtracted from the overall signal? More details on the experimental procedure would be useful;
 - Figure 1F shows two panels, one reporting the SPR sensorgram from WT NOX4 (upper panel) and one reporting the SPR sensorgram from mutant NOX4 (bottom panel). In both of them there is a very small SPR signal (positive in the former and negative in the latter) when ATP is flowed in the cell. No dissociation phase seems to appear and the overall signal is very low in quality, not entirely justified by the ATP small molecular weight. In the case of a negative SPR signal, ATP-protein interaction cannot be excluded. On the contrary, the reasons for a negative SPR signal should be interpreted. Many things could be responsible for such trend, including pH, ionic strength, conformational changes, surface molecules rearrangement.
- In my opinion the SPR data reported from the author are NOT a proof of a different ATP binding to the two proteins, while a much better experimental approach needs to be performed if any conclusions have to be drawn from the SPR data. Replicates at different conditions and a stronger experimental evidence are very much needed in my opinion.

Specific Comments to Reviewers.

We sincerely thank the reviewers for their supportive comments and insightful suggestions. We have comprehensively revised the manuscript which includes new data requested by the reviewers and have concisely restructured the figure legends and detailed methods for clarity. We have undertaken numerous approaches with scientific rigor *in vitro*, *in vivo*, and *ex vivo* to demonstrate scientific reproducibility of our findings. We hope the reviewers' share our enthusiasm that the overall revised manuscript has been strengthened by these changes, which are highlighted in **BOLD**.

Point by point comments to each reviewers question is answered below.

Reviewer 1. We thank the reviewer for the comments that the manuscript is....interesting with novel mechanisms being identified. Reviewer 1 felt that the *in vivo* data supported the conclusions drawn by the authors as opposed with some noted concerns regarding *in vitro* data.

A. In general, the figure legends and materials and methods, were not sufficiently detailed including time of treatment, concentration of chemicals used etc.... We apologize for this oversight and agree with the reviewer that reproducibility within the scientific community is very important. The senior author assures the reviewer that each experiment was approached with scientific rigor and reproducibility. The methods and figure legends have been extensively revised as suggested by the reviewer for clarity.

With regards to appropriate controls, the reviewer raised some concerns:

- **shControl for apoptotic assay in Fig. 6c and 6d.** We had already demonstrated under multiple occasions that shVector controls were resistant to drug-induced apoptosis and sh-mediated knockdown of NOX4 (shNOX4) sensitized this effect (Figs 4A, 4B, 4F, 5A, S7, S9A, and S9B). For this particular experiment, we were asking the question if the lysosomal inhibitor, 3-Methyladenine (3-MA) or genetic silencing of PCAF, or p300 would impede drug-induced apoptosis in shNOX4 stable knock-down RCC clones.
- **Fig. 4f loading control.** We have included Actin as a loading control.
- **Fig. 2f loading control.** We immunoprecipitated NOX4 using equal concentrations of mitochondrial fractions (4-independent aliquots each at 250ug) or IgG control. The Western blot of NOX4 from the immunoprecipitation was to demonstrate NOX4 was present in each tube that was being analyzed in the NOX activity assay. We have clarified our methods and data interpretation in the revised manuscript.
- **Data not shown.** Per the reviewers suggestion, we have included or removed any data not shown in the revised manuscript.

B. Specific notes. In figure 1 how are negative ATP incorporation measurements possible?

The data is presented as dpm to bkg (background). The background for each experiment varied based on washing efficiency and therefore some blanks showed higher background compared to the membranes which were spotted with recombinant protein, but did not bind radioactive ATP. If the reviewer requests, we can subtract the background and present the data as dpm (minus background). This would zero out the negative numbers. For now, we have left the result as is and discussed the background as it contributes to the negative number in the revised manuscript.

C. Figure 2. Mitotracker red more cells should be shown. We have included more cells as per the reviewers suggestion, now presented in Supplementary figure 2a,b,c.

D. LDH can be found in the mitochondria depending on the isoform. We used anti-Lactate Dehydrogenase antibody (abcam ab55433). This mouse monoclonal antibody recognizes the cytoplasm form of LDH only as our mito/cyto fraction clearly demonstrates the mitochondrial fraction is negative for LDH reactivity. We have clearly detailed the antibody in the revised methods section of the manuscript.

- E. Figure 3, panel A. A marker of mitochondria and potential should be used as loss of staining could be through loss of mitochondrial weight or change in membrane potential, which affects mitoSOX uptake.** We did not measure mitochondrial membrane potential in this study; however, we are confident that our measurements using mitoSOX are not altered by changes in mitochondrial membrane potential as the literature suggests the presence or absence of VHL does not change mitochondria membrane potential (Hervouet et al., 2005) and we have previously published that NOX4 knockdown does not alter membrane potential when examining mitoSOX (Block PNAS 2009, Gorin JBC 2013). Moreover, we assessed ROS within this compartment using other methods which support the results.
- F. Panel E, VHL flag should be shown for both cell lines.** We have included FLAG expression in A498 cells as requested. Please see new Supplementary Fig. 5f.
- G. Figure 4.** In figure 4a, we present the data as % apoptosis compared to the buffer treated control. In Figure 4c, we present the data as fold change. We typically see some basal level of apoptosis in drug-treated VHL-deficient cells (1-3 fold), depending on the duration of treatment where our treatment times typically fell between 12-14 hours or if the cells were transiently transfected (which usually added stress to the cells) prior to drug exposure. Due to these variations, the results should be compared within the experiment itself where the aforementioned conditions are the same within that experiment. In experiment 4c, we do detect basal etoposide effects; the importance of this finding is that quenching mitochondrial superoxide, by mitotempol, further sensitizes the VHL-deficient to drug-induced cell death by 2-fold. Together this supports ROS within the mitochondria as playing a role in RCC drug-resistance.
- H. Figure 6a. The reviewer points out that the acetylation band of PKM2 is not clear.** We have indicated the band in the Western blot by using an arrow and have clarified the text by describing the predicted molecular weight of migration (55-60)kDA.
- I. Figure 7. The reviewer suggests that changes of cell growth between shVec and shNOX4 clones, in lieu of cell apoptosis, may be a reason for the changes observed.** We examined the effects of NOX4 and PKM2 knockdown on cell growth using thymidine incorporation assay (routinely done in our laboratory). We find that stable or transient knockdown of NOX4 or PKM2 respectively did not reduce, or alter, cell growth in VHL-deficient RCC cells (new Figure S10a and S10b respectively). Together, this supports our observation that NOX4 silencing sensitizes VHL-deficient cells to drug-induced cells death.
- J. The reviewer also indicates that continued apoptosis after 4-days of stopping etoposide *in vivo* was surprising.** We show that cell death is evident in the shNOX4 tumors compared to the shVector controls in the same mouse as detected by cleaved caspase 3 (IHC) and cleaved PARP (western). The general half-life of etoposide *in vitro* is relatively short as the reviewer points out; however, etoposide does have prolonged tissue storage. If the etoposide remains in the tissue and is active for longer duration (days) or if the PARP cleavage and Caspase-3 cleavage products are stable enough to be detected after 4 days (more likely) would be hard to determine.
- K. Figure S2. The reviewer suggests the data are consistent with a protein that is on the inside of the outer mitochondrial membrane.** We completely agree with the reviewer and have changed the wording in the revised manuscript which now reads: Together our data suggests that NOX4 resides in the inner mitochondrial membrane **or on the inside of the outer mitochondrial membrane.**

Reviewer 2. We appreciate the comments from Reviewer 2 that the study is interesting, original, and of importance and that the experimental work is rigorous and well done. However, Reviewer 2 did note concerns to be addressed.

- A. The reviewer stated that NADPH-dependent superoxide and H₂O₂ generation should be compared in the NOX4 ATP mutant compared to the NOX4 WT.** As requested by the

reviewer, we evaluated NADPH oxidase activity to complement the mitoSOX data. Please see new Fig. 4f.

- B. The reviewer points out that other laboratories have identified NOX4 isoforms within the nucleus and endoplasmic reticulum and suggests we compare the NOX4 within the mitochondria to these compartmental organelles.** We are aware of these data, but feel this is outside the scope of this manuscript. We used mitotempol to demonstrate that blocking superoxide within the mitochondrial compartment is necessary and sufficient to drive drug-resistance in VHL-deficient RCC cells. We have, however, brought NOX4 localization as a point of discussion in the revised manuscript.
- C. The reviewer states that our suggestion that p22phox may not localize to the mitochondria is an important and interesting finding and requests that we examine the expression of p22phox in our mitochondrial fractions and other organelle fractions.** We agree with the reviewer that expression and function of p22phox is interesting when considering NOX4 localization to the mitochondria, but with the manuscript already highlighting such a breadth and depth of novel mechanisms, we did not want to divert into all organelles and cellular compartments. We do agree with the reviewer however and have examined p22phox's localization in the mitochondria vs cytosolic fractions and indeed show that p22phox does not harbor sequences which may target it to the mitochondria or localize to this organelle in RCC cells. The new data is shown Supplementary figure 11 a, b.
- D. The reviewer requests that the authors better clarify cellular fractionation as diluted proteins may not detect expression.** We agree and have thought about these factors during our approach. We loaded all fractions using equal protein (typically 30-50ug) to avoid this problem. Moreover, we probed for known proteins within these fractions to validate purity of our fractionation technique.
- E. Figure S2, in general there are mistakes....the supplemental data in the manuscript and the corresponding numbers are wrong and legends are missing.** We sincerely apologize and acknowledge other reviewers also requested clarity in figure legends, methods, and figures. We have carefully revised the manuscript to clarify the aforementioned issues raised by the reviewers.
- F. The reviewer noted the more abundant reduction of superoxide than hydrogen peroxide reduction in shNOX4 cells and ATP effects.** We agree with the reviewer that silencing of NOX4 or ATP-inhibitable of NOX4-derived superoxide is more robust than hydrogen peroxide. This suggests that either NOX4 does not contribute to the majority of hydrogen peroxide detected with the cells, but can also be explained by the complexity of measuring of ROS as detection of ROS can be dependent on ROS species interacting with other intracellular ROS species in addition to the antioxidants present to consume these species etc... Additionally, we still don't know much about how NOX4 regulates its ROS generation within the mitochondrial compartment, which is why our findings of ATP regulating the enzyme and the absence of p22phox so important.
- G. Figure 3. A western blot should show in A498 cells that adding back VHL does not change NOX4 expression.** We completely agree with the reviewer. We now provide a Western blot of NOX4 in A498 cells with VHL added back.
- H. The authors should discuss the fact that NOX4 inhibition changes ETC in HUVEC cells...** Agree. We have now included this in the discussion.
- I. 4C. How long are the cells treated with MitoTempol?** 6-hrs pretreatment mitotempol then 12-14 with etoposide. Clarified in the figure legend and in methods.
- J. Please provide PKM2 expression for 6C and 6D.** We have included PKM2 expression in this panel (new figure 6e) as requested by the reviewer.
- K. Please provide IHC or western for PKM2 in vivo animal tissue.** We have included PKM2 expression in this panel (new figure 7b) as requested by the reviewer.

- L. **Figure 8, please provide a western for NOX4 and PKM2 in ex vivo cells.** We have included PKM2 (new figure 8h, i) and NOX4 (new figure 8j, k) expression in this panel as requested by the reviewer.
- M. **Effect of shVector and shNOX4 on PCAF expression.** The reviewer has raised an interesting point. We did not look at PCAF expression in the shNOX4 cells, as we felt the manuscript was complex as is and elucidating the mechanism by which NOX4 regulates PCAF-dependent regulation of PKM3 K305 acetylation is outside the scope of this study. We have included a discussion on this point in the revised manuscript.

Reviewer 3. We thank the reviewer for the comments that the manuscript is well-written laying out a series of elegant step-wise experiments that clearly define novel roles and mechanisms by which NOX4 mediates drug-resistance in RCC cells, which represent key contributions important for cancer therapeutics. They hypothesis and model are clearly described and figures clear and compelling.

Minor comments:

- A. **Introduction, paragraph 4 should read, alternative splicing.** Corrected, Thank-you.
- B. **Introduction paragraph 5: first sentence should read, “and has the highest death rate”.** Corrected, Thank-you. **Suggest that the following object be added, “despite surgery to remove the affected kidney, ~30-40% of patients succumb to metastatic disease.** Corrected, Thank-you.
- C. **Could the authors comment on NOX4 in the ER? Comment on discrepancy.** We do not feel that NOX4 localization to the ER or nucleus in other cell types is a discrepancy to our findings herein. We know that alternative splice variants of NOX4 have been detected and that NOX4 may have different localizations in different cell types and each variant may respond to different agonists. We did not look in the ER fraction for NOX4 in this study as we did not want to distract the manuscript toward these complex issues, which we have discussed in comprehensive reviews (Block Nat Cancer Review 2012). However, we do feel the various localizations of NOX4 should be discussed and have included a discussion on this in the revised manuscript.

Reviewer 4. SPR Technical, notes to authors:

- A. **It should be Biacore, not Biocore.** We have changed the wording, which now reads: As an independent approach, we evaluated WTNOX4 and mutNOX4 binding affinity to ATP using surface plasmon resonance-based technology (Pioneer SensiQ, Sinsiq Technologies, performed by Affina Biotechnologies).
- B. **More details on the experimental procedure would be helpful. Authors state that “equal concentration of WT NOX4 (341-561) and mutant NOX4 (341-561 K540A)” were used: what was the concentration? Have the authors any ideas on surface coverage? What was the amount of immobilized proteins on the surface?** Based on the reviewers suggestion, we have asked Affina Biotechnology for updated the details of the experiments and updated this in Methods, which now incorporate the following: NOX4 (both WT and mutant) were immobilized on a HisCap chip (SensiQ Technologies). For immobilization the HisCap chip was charged with Ni and activated with 0.05M NHS/0.2 M EDC for 3 min. The protein was injected at 50ug/mL on the activated surface to achieve the maximal loading of 1300RU for both wild-type and mutant proteins after which the surface was deactivated with 1 M ethanolamine. Immobilization was performed in HBS-P at 25C, while binding was tested at 4C in the binding buffer as above. The reference surface was activated and deactivated in the same manner and did not contain any protein.
- C. **Was the reference cell response subtracted from the overall signal? More details on the experimental procedure would be useful; For all analyses the double reference method was used.** Samples were injected in a FastStep injection mode (SensiQ Tehcnologies) resulting in a sequential injection of 1.56, 3.13, 6.25, 12.5, 25, 50, 100 uM ATP followed by a

single dissociation step.

- D. Figure 1F shows two panels, one reporting the SPR sensorgram from WT NOX4 (upper panel) and one reporting the SPR sensorgram from mutant NOX4 (bottom panel). In both of them there is a very small SPR signal (positive in the former and negative in the latter) when ATP is flowed in the cell. No dissociation phase seems to appear and the overall signal is very low in quality, not entirely justified by the ATP small molecular weight. In the case of a negative SPR signal, ATP-protein interaction cannot be excluded. On the contrary, the reasons for a negative SPR signal should be interpreted. Many things could be responsible for such trend, including pH, ionic strength, conformational changes, surface molecules rearrangement. It is likely that components of the injection sample produced the negative binding response of the mutant as the result of small differences in the surfaces of the specific and reference channels not compensated by the specific binding of ATP to the NOX4. While not sufficient to prove that the mutant NOX4 on its own it is consistent with other binding studies performed on wild-type and mutant proteins.**
- E. In my opinion the SPR data reported from the author are NOT a proof of a different ATP binding to the two proteins, while a much better experimental approach needs to be performed if any conclusions have to be drawn from the SPR data. Replicates at different conditions and a stronger experimental evidence are very much needed in my opinion.** We understand this reviewer did not read the manuscript in its entirety and therefore did not see the additional data we presented conducting exhaustive analysis of *in vitro* ATP binding assays, titrations with cold ATP and mutant NOX4. We actually did SPR as an independent approach to support the extensive *in vitro* binding data.

Reviewers' comments:

Reviewer #1 (Remarks to the Author):

The authors have addressed many of the issues raised by the reviewers, and the manuscript is much improved as a result. However, they appear to have missed some important points in their rebuttal, which were also not addressed in the manuscript.

Specific points raised that were not addressed thus far:

1. Supplementary Figure 2 - although more cells are shown, the quality of the mitotracker red remains poor. The non-specific red particles are particularly troublesome and new images need to be acquired. The same is true for Figure 2A.
2. Figure 4. In their rebuttal, the authors point out that basal apoptosis can be observed in VHL-negative cells, and that siRNA can result in changes to the response to etoposide. They also comment that results should be compared within the experiments themselves. Although this second point has some truth, reproducibility between experiments is critical, and although the magnitude of a response could be expected to vary, the nature of it should not. The difference between the first three conditions in panel a, and the first two in panel c are the presence of a stably-expressed control shVector, which should not have a significant effect on the biological outcomes of etoposide treatment. However, in panel a, etoposide fails to induce any cell death. In panel c, around 3-fold increase in apoptosis is observed. These data suggest that the shVector has a biological effect, which therefore throws doubt on the experiments performed in panel a in their entirety.
3. Generation of knockdown lines - the sequences of the shRNA vectors used are still not shown.
4. Concentrations of inhibitors used in the respiratory experiments are still lacking in the materials and methods
5. It is still not clear how 'apoptosis' has been assessed when using the annexin V/PI method - just the PI & Annexin V position, or including the PI negative (i.e. early apoptosis) fraction?

Reviewer #2 (Remarks to the Author):

The authors replied satisfactorily to my questions.

This paper is of a great importance and identifies a new role of NOX4 as a mitochondrial ROS-generating system involved in chemo resistance via its inhibition of PKM2 degradation

Reviewer #3 (Remarks to the Author):

The edited manuscript addresses all concerns of this reviewer and has been improved by the additions. No further concerns or recommendations.

Reviewer #4 (Remarks to the Author):

As pointed out by the authors, I was questioning only the validity of the SPR data, not the overall quality of the paper. The authors ignored my comments about the validity of the reported SPR data and in my opinion the latter do not add any additional evidence of a difference between ATP binding to recombinant WTNOX4 and the MUTNOX4. If the other reviewers consider the overall

work valuable without the SPR data, then I would rather have the SPR data removed altogether. In my opinion, it is not the quantity of the data that counts but their quality, so I cannot see any reasons for showing data that do not prove anything at all. The reported SPR data are faulty.

Specific Comments to Reviewers.

Please find our point-by-point comments to the points raised by the reviewers in the revised manuscript below. We hope we have appropriately addressed all the reviewers' questions/concerns. The new data and text have been added to the revised manuscript in **BOLD**.

Reviewer #1:

The authors have addressed many of the issues raised by the reviewers, and the manuscript is much improved as a result. However, they appear to have missed some important points in their rebuttal, which were also not addressed in the manuscript... Apologies. Please see below.

1. Supplementary Figure 2 - although more cells are shown, the quality of the mitotracker red remains poor. The non-specific red particles are particularly troublesome and new images need to be acquired. The same is true for Figure 2A. Agree. We have repeated these experiments to rid of the MTR spotted background. The new data are included in the revised manuscript.

2. Figure 4. In their rebuttal, the authors point out that basal apoptosis can be observed in VHL-negative cells, and that siRNA can result in changes to the response to etoposide. They also comment that results should be compared within the experiments themselves. Although this second point has some truth, reproducibility between experiments is critical, and although the magnitude of a response could be expected to vary, the nature of it should not. The difference between the first three conditions in panel a, and the first two in panel c are the presence of a stably-expressed control shVector, which should not have a significant effect on the biological outcomes of etoposide treatment. However, in panel a, etoposide fails to induce any cell death. In panel c, around 3-fold increase in apoptosis is observed. These data suggest that the shVector has a biological effect, which therefore throws doubt on the experiments performed in panel a in their entirety. We have looked at the data and understand the comments made by the reviewer and would like to clarify. The reviewer states that the first three conditions in panel 4a and the first two panels of 4c are the same when in fact they are different. In panel 4a, we used the shRNA stable knockdown clones (shVEC) as a control for (shNOX4). On the other hand, panel 4c is 786-O cells with no stable transfection. Moreover, although the VHL-deficient 786-O cell line did demonstrate etoposide-induced apoptosis compared to buffer control in Figure 4 panel C, the inclusion of mitotempol significantly sensitized the cells to etoposide-induced apoptosis compared to etoposide/buffer control alone. We have included this extra analysis in the revised manuscript for clarity.

We agree with the reviewer that assays should demonstrate reproducibility and continuity under the same conditions. We would like to ensure the reviewer that we have performed several independent experiments with an n of 3 independent plates each time for each condition (within this study and others not related to this study) and are confident that our control panels are consistent. We thank the reviewer for this discussion.

3. Generation of knockdown lines - the sequences of the shRNA vectors used are still not shown. We thank the reviewer for bringing this oversight to our attention. We have included the sequences in the materials and methods. Briefly: Mission Lentiviral transduction particles, (SHCLNV product#; NM_016931 geneID; Human): (Sigma Aldrich): NM_016931 TRCN0000046089 (Sequence: CCGGGCTGTATATTGATGGTCCTTTCTCGAGAAAGGACCATCAATATACAGCTTTTTG) and TRCN0000046090 (Sequence:CCGGCCCTCAACTTCTCAGTGAATTCTCGAGAATTCAGTACTGAGAAGTTGAGGGTTTTG).

4. Concentrations of inhibitors used in the respiratory experiments are still lacking in the

materials and methods. For Seahorse experiments, we measured basal oxygen consumption. We have clarified this in the revised manuscript. However, as the reviewer pointed out, we overlooked to include the concentration and incubation time for Rotenone in the materials and methods for ATP experiments, which have now been included in the materials and methods of the revised manuscript. Briefly, for experiments using rotenone, specified cells were treated with rotenone (1 μ M) for 30 minutes.

5. It is still not clear how 'apoptosis' has been assessed when using the annexin V/PI method - just the PI & Annexin V position, or including the PI negative (i.e. early apoptosis) fraction? For Annexin V and PI analysis, we have included both early (Annexin V positive;PI negative) and late (Annexin V positive;PI positive) apoptosis. We have integrated this into the materials and methods section of the revised manuscript.

Reviewer #2:

The authors replied satisfactorily to my questions and this paper is of a great importance and identifies a new role of NOX4 as a mitochondrial ROS-generating system involved in chemo resistance via its inhibition of PKM2 degradation. We thank the reviewer for the positive feedback and support. The reviewers comments throughout this process have been instrumental in strengthening the overall manuscript.

Reviewer #3:

The edited manuscript addresses all concerns of this reviewer and has been improved by the additions. No further concerns or recommendations. We thank the reviewer for the positive feedback and support. The reviewers comments throughout this process have been instrumental in strengthening the overall manuscript.

Reviewer #4:

As pointed out by the authors, I was questioning only the validity of the SPR data, not the overall quality of the paper.... If the other reviewers consider the overall work valuable without the SPR data, then I would rather have the SPR data removed altogether... The reported SPR data are faulty. We apologize for any misunderstanding. We understand the reviewer is providing expert advice, which we sincerely appreciate. We have discussed the data with the President/CEO of Affina Biotechnology who has underscored to us that the expected R_{max} for ATP binding to 1300RU of NOX4 is ~8RU. Thus we get ~50% of the expected R_{max}. Considering that some protein may be inactivated during immobilization that uses covalent modification of lysines such a loss of activity does not seem unreasonable. Moreover, the starting material may only have 50% of the active protein to begin with - that is not unusual in purifications.

It should be noted that we do not have the resources to spend thousands of more dollars doing more extensive SPR assays, which may or may not yield higher expected RU values. We are confident concluding that the SPR data provides qualitative support to our *in vitro* binding assays (which include titration biological and function outputs, cold ATP competition experiments, and site directed mutant Walker ATP binding domain of NOX4), which together demonstrate that NOX4 directly binds ATP directly through the Walker A binding motif.

If our expert SPR reviewer remains resolute that the data should be removed, and the other reviewers agree, we will remove the data as requested. On the other hand, if the reviewer would like us to include some specific language in the manuscript to allow the data to be included, we are also supportive to that. We have kept the data in the revised manuscript until the reviewers have advised.

REVIEWERS' COMMENTS:

Reviewer #1 (Remarks to the Author):

The authors have now responded to all the comments I have made, and I believe that the manuscript is now much improved. The manuscript is of great interest to the field, and with the quality of data, is likely to be of significant impact.

Reviewer #4 (Remarks to the Author):

In order to remove the SPR artefact in Fig. 1f, bottom (a negative SPR signal), some additional experiments could be carried out as explained, for example, at <https://www.sprpages.nl/experiments/troubleshooting>.

However, as the authors have clearly expressed their impossibility to perform additional experiments and considering that in this case the SPR data are only to support some other experimental evidences (and I can understand that), I would suggest to modify the text as follows:

"As an independent approach, we evaluated rNOX4341-561 and rNOX4341-561MUT binding affinity to ATP using surface plasmon resonance-based technology (Pioneer SensiQ, Sinsiq Technologies, performed by Affina Biotechnologies). Equal concentrations of rNOX4341-561 and rNOX4341-561MUT proteins were immobilized by covalent amine-directed immobilization on a Ni-NTA chip surface using the Ni-His tag interaction as a pre-condensation step. The results show a rise in signal (RU) peaking at 80-100s for rNOX4341-561 when ATP was passed over the flow channel whereas rNOX4341-561MUT showed no binding/interaction signal (Fig. 1f top and bottom panels respectively).

Unfortunately, a negative SPR signal is present in Fig1f, bottom. Although it is not easy to identify the exact cause of such experimental artefact (buffer mismatch, volume exclusion, non-specific matrix interaction and/or non-specific reference interaction), it is clear that there is an important difference in the interaction of rNOX4341-561 and rNOX4341-561MUT with ATP. Therefore, although in a very qualitative manner, our SPR data show that NOX4 directly binds ATP directly through the Walker A binding motif."